# Deployment Efficient Reward-Free Exploration with Linear Function Approximation

## Abstract

We study deployment efficient reward-free exploration with linear function approximation, where the goal is to explore a linear Markov Decision Process (MDP) without revealing the reward function, while minimizing the number of exploration policies used during the algorithm. We design a new reinforcement learning (RL) algorithm whose sample complexity is polynomial in the feature dimension and horizon length, while achieving near-optimal deployment efficiency for linear MDPs under the reward-free exploration setting. More specifically, our algorithm explores a linear MDP in the reward-free manner, while using at most $H$ exploration policies during its execution where $H$ is the horizon length. Compared to previous algorithms with similar deployment efficiency guarantees, the sample complexity of our algorithm does not depend on the reachability coefficient or the explorability coefficient of the underlying MDP, which can be arbitrarily small for certain MDPs. Our result addresses an open problem proposed in prior work. To achieve such a result, we show how to truncate state-action pairs of the underlying linear MDP in a data-dependent manner, and devise efficient offline policy evaluation and offline policy optimization algorithms in the truncated linear MDP. We further show how to implement reward-free exploration mechanisms in the linear function approximation setting by carefully combines these offline RL algorithms without sacrificing the deployment efficiency.

## 1 Introduction

In real-world reinforcement learning applications, deploying new policies usually comes at a cost. For instance, in robotics applications (Kober et al., 2013), deploying new policies involves operations on the hardware level, which typically requires long waiting periods. As another example, in medical applications (Almirall et al., 2012; 2014; Lei et al., 2012), it is unrealistic to deployment new policies frequently, since switching to a new policy typically requires a separate approval process which usually involves domain experts and could therefore be costly. In recommendation systems (Theocharous et al., 2015), the deployment of a new policy often takes weeks, as the new recommendation strategy must pass internal tests to ensure safety and practicality before being deployed, which again can be time-consuming. On the other hand, in all these scenarios, although switching the policy based on instantaneous data (as required by typical RL algorithms) is infeasible, once a policy is deployed, it is possible run a large batch of experiments in parallel to collect new data. Therefore, in such applications, the agent needs learning a good policy while minimizing the number of policy deployments.

Empirically, the notion of *deployment efficiency* was first proposed by Matsushima et al. (2020), while formal definition of deployment complexity was recently defined by Huang et al. (2022). Intuitively, deployment complexity measures the total number of policy deployments of a RL algorithm, while requiring the interval between policy switching, i.e., the number of trajectories collected before switching to a new policy, is fixed in advance. Under the notion of deployment complexity, a line of recent work designed provably efficient RL algorithms (Huang et al., 2022; Qiao et al., 2022; Qiao & Wang, 2022) in various settings. In particular, for the tabular setting where the state space is assumed to be discrete and have small size, Qiao et al. (2022) designed a provably efficient RL algorithm with $O(H)$ policy deployments. Huang et al. (2022); Qiao & Wang (2022) studied the deployment complexity of RL with linear function approximation. In particular, in the linear MDP (Yang & Wang, 2019; Jin et al., 2023) setting, the sample complexity of the algorithms

by Huang et al. (2022); Qiao & Wang (2022) is polynomial in the feature dimension $d$ and the horizon length $H$, while the deployment complexity is $O(dH)$ or $O(H)$. Moreover, it has been shown in Huang et al. (2022) that any RL algorithm for linear MDPs requires a deployment complexity of $\tilde{\Omega}(H)$[1].

Although prior works mentioned above seem to give a complete answer to the deployment complexity of RL for linear MDPs, it turns out that to achieve the nearly optimal $O(H)$ deployment complexity, existing algorithms either work in the tabular setting (Qiao et al., 2022) and therefore cannot handle cases where state space is enormous or continuous, or require additional reachability assumption (Huang et al., 2022) or explorability assumption (Qiao & Wang, 2022) which, roughly speaking, assumes that all directions of the feature space can be explored by some policy. Such reachability assumption and explorability assumption could be quite restrictive and would significantly limit the scope that the RL algorithms can be used. In these assumptions, it is usually assumed that some type of "reachability coefficient" is lower bounded, and the sample complexity of existing algorithms with $O(H)$ deployment complexity all have polynomial dependency on the reciprocal of the reachability coefficient. In the tabular setting, assuming the reachability coefficient is lower bounded is equivalent to assuming all states in the state space can be reached by some policy with lower bounded probability, and for a general linear MDP, such reachability coefficient could be arbitrarily small in which case the sample complexity of existing algorithms with $O(H)$ deployment complexity would be infinite. In order to give a satisfying answer to the deployment complexity of RL in linear MDPs, in this paper, we study the following question:

*Is is possible to design RL algorithms for linear MDPs with nearly optimal deployment complexity and polynomial sample complexity, without relying on any additional assumptions?*

In fact, such a question was mentioned explicitly in prior work (Huang et al., 2022; Qiao & Wang, 2022) and was left as an important direction for future investigation. In particular, it was conjectured in Huang et al. (2022) that to achieve $O(H)$ deployment complexity, relying on additional assumptions like reachability or explorability is unavoidable.

**Our Contribution.** In this paper, we resolve the question mentioned above by designing a new RL algorithm for linear MDPs with $H$ deployment complexity. Our new algorithm achieves polynomial sample complexity for any linear MDP and does not rely on additional assumptions. In fact, our new algorithm works for the reward-free exploration setting (Jin et al., 2020; Wang et al., 2020a; Chen et al., 2022; Wagenmaker et al., 2022; Zhang et al., 2021b) and does not require access to the reward distribution during its exploration phase, giving it additional favorable properties that could be beneficial for practical use. The formal guarantee of our new algorithm is informally summarized in the following theorem.

**Theorem 1** (Informal version of Theorem 4). *For reward-free exploration in linear MDPs, there exists an algorithm (Algorithm 1) with deployment complexity $H$, such that with probability $1-\delta$, the algorithm returns a policy whose suboptimality is at most $\epsilon$, whose sample complexity is polynomial in $d$, $H$, $1/\epsilon$ and $\log(1/\delta)$. Here, $d$ is the feature dimension and $H$ is the horizon length.*

Combined with existing hardness result (Huang et al., 2022), our new result in Theorem 1 gives a complete answer to the deployment complexity of RL for linear MDPs, and shows that additional assumptions like reachability or explorability conjectured to be unavoidable in previous work, are in fact not necessary for achieving a nearly optimal deployment complexity.

The remaining part of this paper is organized as follows. Section 2 give an overview of related work. Section 3 introduces necessary technical backgrounds and notations. Section 4 gives an overview of our new technical ideas. Section 5 and Section 6 introduce the formal definition of our algorithms together with an overview of its analysis. Most of the proofs are deferred to the supplementary material.

---

[1]Throughout this paper, we use $\tilde{O}$ and $\tilde{\Omega}$ to suppress logarithmic factors.

## 2 RELATED WORK

There is a large and growing body of literature on the sample complexity of reinforcement learning. We refer interested readers to the monograph by Agarwal et al. (2019) for a more thorough review, and focus on most relevant work in this section.

**Deployment Efficiency and Other Notions of Adaptivity.** The notion of *deployment efficiency* was first proposed in the empirical work by Matsushima et al. (2020), while its formal definition was first defined by Huang et al. (2022). Under this notion, Huang et al. (2022); Qiao et al. (2022); Qiao & Wang (2022) designed provably efficient RL algorithms in various settings. As mentioned in the introduction, in order to achieve a nearly optimal deployment complexity, existing algorithms either work in the tabular setting, or rely on additional reachability assumption or explorability assumption which we strive to avoid in this work. Zhao et al. (2023) designed deployment efficient RL algorithms for function classes with bounded eluder dimension. However, even for linear functions, the deployment complexity of the algorithm by Zhao et al. (2023) is $\tilde{O}(dH)$ is far from being optimal.

The notion of deployment efficiency is closely related to the low switching setting (Bai et al., 2019; Zhang et al., 2020c; Gao et al., 2021; Kong et al., 2021; Qiao et al., 2022; Wang et al., 2021). We refer readers to prior work (Huang et al., 2022; Qiao et al., 2022) for a detailed comparison between these two different notions. Roughly speaking, in the low switching setting, the agent decides whether to update the policy after collecting each trajectory. On the other hand, the notion of deployment efficiency requires the interval between policy switching to be fixed, and therefore, deployment efficient RL algorithms are easier to implement in practical scenarios. The low switching setting was also studied for other sequential decision-making problems including bandits problems (Abbasi-Yadkori et al., 2011; Cesa-Bianchi et al., 2013; Simchi-Levi & Xu, 2019; Ruan et al., 2021).

**Reward-free Exploration.** The notion of reward-free exploration was first proposed by Jin et al. (2020). In this setting, the agent first collects trajectories from an unknown environment without any pre-specified reward function. After that, a specific reward function is given, and the goal is to use samples collected during the exploration phase to output a near-optimal policy for the given reward function. The sample complexity of reward-free exploration was studied and improved in a line of work (Kaufmann et al., 2021; Ménard et al., 2021; Zhang et al., 2020b) A similar notion called task-agnostic exploration was consider by Zhang et al. (2020a). For linear MDPs, the first polynomial sample complexity for reward-free exploration was obtained by Wang et al. (2020a). Later, the sample complexity was improved by Zanette et al. (2020); Wagenmaker et al. (2022). Reward-free exploration was also considered in other RL settings including linear mixture MDPs (Chen et al., 2022; Zhang et al., 2021a) and RL with non-linear function approximation (Chen et al., 2022).

**Technical Comparison with Existing Algorithms.** Finally, we compare our new algorithm with existing algorithms with $O(H)$ deployment complexity (Qiao et al., 2022; Qiao & Wang, 2022) from a technical point of view, and a more detailed overview of our new technical ingredients is given in Section 4. To achieve a nearly optimal $O(H)$ deployment complexity in the tabular setting, Qiao et al. (2022) applied absorbing MDP to ignore those "hard to visit" states. In this work, similar ideas are used, though we work in the linear MDP setting which is much more complicated than the tabular setting and therefore requires a more careful treatment. In order to design an algorithm with $O(H)$ deployment complexity in linear MDPs under the explorability assumption, Qiao & Wang (2022) showed how to solve a variant of G-optimal experiment design in an offline manner. In this work, we also use offline policy optimization and offline policy evaluation to build exploration policies in linear MDPs. However, the lack of the explorability assumption raises substantial more technical challenges which necessitates more involved algorithms and analysis.

## 3 PRELIMINARIES

In this section, we introduce the basics of MDPs, the learning problem and our assumptions. We use $\Delta(X)$ to denote the set of probability distributions over the set $X$, and $[N]$ to denote the set $\{1, 2, \ldots, N\}$ for a positive integer $N$.

**Episodic MDPs.** A finite-horizon episodic MDP can be characterized by a tuple $(\mathcal{S}, \mathcal{A}, R, P, H, d_{\text{ini}})$, where $\mathcal{S} \times \mathcal{A}$ denotes the state-action space, $R : \mathcal{S} \times \mathcal{A} \times [H] \to \Delta([0,1])$ is the reward distribution (with mean $r := \mathbb{E}[R]$), $P : \mathcal{S} \times \mathcal{A} \times [H] \to \Delta(\mathcal{S})$ is the probability transition kernel, $H$ is the planning horizon and $d_{\text{ini}} \in \Delta(\mathcal{S})$ is the initial distribution.

Moreover, a policy $\pi = \{\pi_h : \mathcal{S} \to \Delta(\mathcal{A})\}_{h=1}^{H}$ is a group of mappings from the state space $\mathcal{S}$ to the distributions over $\mathcal{A}$. We say $\pi$ is a deterministic policy if $\pi_h(s)$ is a one-hot vector for all $h$ and $s$. For simplicity, we use $\pi_h(s)$ to denote that action .

In each episode, the learner starts from an initial state $s_1 \sim d_{\text{ini}}$, and then proceeds by observing current state $s_h$, taking action $a_h$ and transiting to $s_{h+1}$ according $P_h(\cdot \mid s_h, a_h)$ for $h = 1, \ldots, H$. Along the trajectory $\{s_h, a_h\}_{h=1}^{H}$, the learner collects reward $\sum_{h=1}^{H} r_h$ where each $r_h$ is drawn according to $R_h(s_h, a_h)$ independently.

Fix a policy $\pi$, we define the $Q$-function and the value function as below:

$$Q_h^\pi(s,a) := \mathbb{E}_\pi \left[ \sum_{h'=h}^{H} r_{h'} \,\Big|\, (s_h, a_h) = (s,a) \right] \quad \text{and} \quad V_h^\pi(s) := \mathbb{E}_\pi \left[ \sum_{h'=h}^{H} r_{h'} \,\Big|\, s_h = s \right]$$

for any $(s,a) \in \mathcal{S} \times \mathcal{A}$ and $h \in [H]$. The optimal $Q$-function and value function at step $h$ can be given as

$$Q_h^*(s,a) = \max_\pi Q_h^\pi(s,a) \quad \text{and} \quad V_h^*(s) = \max_\pi V_h^\pi(s), \quad \forall(s,a) \in \mathcal{S} \times \mathcal{A}, h \in [H].$$

By the Bellman optimality condition, it holds that $V_h^*(s) = \max_a Q_h^*(s,a), \forall s \in \mathcal{S}$, and $Q_h^*(s,a) = r(s,a) + \mathbb{E}_{s' \sim P(\cdot|s,a)}[V_{h+1}^*(s')], \forall(s,a) \in \mathcal{S} \times \mathcal{A}$.

**Linear Function Approximation.** We assume that the transition kernel and the reward function exist within a known low-dimensional subspace, a situation often referred to as a linear MDP.

**Assumption 2** (Linear MDP Jin et al. (2023)). *Let $\{\phi_h(s,a)\}_{(s,a) \in \mathcal{S} \times \mathcal{A}, h \in [H]}$ be a set of known feature vectors such that $\max_{s,a} \|\phi_{s,a}\|_2 \le 1$. For each $h \in [H]$, let $\theta_h \in \mathbb{R}^d$ and $\mu_h \in \mathbb{R}^{S \times d}$ be respectively the reward kernel and transition kernel such that*

$$
\begin{aligned}
r_h(s,a) &= \langle \phi_h(s,a), \theta_h \rangle & &\forall(s,a) \in \mathcal{S} \times \mathcal{A}, \\
P_h(\cdot \mid s,a) &= \mu \phi_h(s,a), & &\forall(s,a) \in \mathcal{S} \times \mathcal{A}, \\
\|\theta_h\|_2 &\le \sqrt{d}, & & \\
\|\mu_h^\top v\|_2 &\le \sqrt{d}, & &\forall v \in \mathbb{R}^S \text{ obeying } \|v\|_\infty \le 1.
\end{aligned}
$$

Under Assumption 2, both the reward function and the transition kernel are linear combinations of a set of $d$-dimensional feature vectors. This allows for effective dimension reduction, provided that $d$ is much smaller than $SA$.

**Reward-free Exploration.** Now we introduce the framework of reward-free exploration. Reward-free exploration comprises two phases: the sampling phase and the planning phase. In the sampling phase, the learner collects a dataset $\mathcal{D}$ by interacting with the environment without reward information, and in the planning phase, given any reward function $\{r_h\}_{h \in [H]}$ satisfying Assumption 2, the learner is asked to output an $\epsilon$-optimal policy with probability at least $1 - \delta$, where $\epsilon$ is a threshold and $\delta$ is the failure probability.

**Deployment-efficient Reward-free Exploration.** We present the definition of deployment complexity for reward-free exploration as follows.

**Definition 3** (Huang et al. (2022)). *We say that an algorithm has a deployment complexity $K$ in linear MDPs if the following holds: given an arbitrary linear MDP under Assumption 2, for arbitrary $\epsilon$ and $\delta \in (0,1)$, the algorithm will conduct $K$ deployments and collect at most $L$ trajectories in each deployment, under the following constraints*

*(a) With probability $1 - \delta$, given any reward kernel $\{\theta_h\}_{h \in [H]}$ satisfying Assumption 2, the learner return an $\epsilon$-optimal policy $\pi$ under this reward kernel, i.e,*
$\mathbb{E}_\pi \left[ \sum_{h=1}^{H} \phi_h^\top(s_h, a_h)\theta_h \right] \ge \max_{\pi'} \mathbb{E}_{\pi'} \left[ \sum_{h=1}^{H} \phi_h^\top(s_h, a_h)\theta_h \right] - \epsilon;$

(b) *The sample size $L$ is polynomial, i.e., $L = \text{poly}(d, H, \frac{1}{\epsilon}, \log(\frac{1}{\delta}))$. Moreover, $L$ should be fixed a priori and cannot change adaptively from deployment to deployment.*

**Notations.** For a symmetric matrix $A$ and a PSD matrix $B$, we write $|A| \preceq B$ iff $B + A \succeq 0$ and $B - A \succeq 0$. Let $\text{Range}_{[a,b]}(x) = \mathbb{I}[x \leq a] \cdot a + \mathbb{I}[a < x < b] \cdot x + \mathbb{I}[x \geq b] \cdot b$ for two reals $a \leq b$. For two PSD matrices $A$ and $B$, define $\text{T}(A, B) := \lambda A$ where $\lambda = \max\{\zeta \leq 1 : \zeta A \preceq B\}$. Define $\theta_h(v) = \mu_h^\top v$ for $v \in \mathbb{R}^{\mathcal{S}}$ and $h \in [H]$. Denote $\mathbf{1}_f$ as the $|\mathcal{S}|$-dimensional one-hot vector with element 1 in the dimension of $s$. We use $\Pr[\cdot]$ to denote the probability of an event.

## 4 TECHNICAL OVERVIEW

In this section, we give an overview of the technical challenges behind achieving Theorem 1, together our new ideas for tackling these challenges.

**The Layer-by-layer Approach.** Similar to existing algorithms with $O(H)$ deployment complexity (Huang et al., 2022; Qiao et al., 2022; Qiao & Wang, 2022), our new algorithm is based on a layer-by-layer approach. For each layer $1 \leq h \leq H$, based on an offline dataset obtained during previous iterations, our algorithm designs a exploration policy (a mixture of deterministic policies) for layer $h$, collect an offline dataset using the exploration policy, and then proceed to the next layer $h + 1$ inductively. Since we only use a single exploration policy for each layer, and there are $H$ layers, the deployment complexity of such an approach would consequently be $H$. Following such an approach, datasets obtained for previous layers will be used for the purpose of policy optimization and policy evaluation for later layers, and therefore, the dataset should be able to cover all directions in the feature space. Therefore, we must carefully design the exploration strategy, so that for any direction that can be reached by some policy, our exploration strategy could also reach that direction up to an appropriate competitive ratio. By repeatedly sample trajectories by following the exploration strategy, we would get a dataset that would be sufficient for the purpose of policy optimization and policy evaluation for later layers

**Dealing with Infrequent Directions.** The main technical issue associated with the approach mentioned above, is that there could directions that cannot be reached frequently by any policy. In such a case, it is unrealistic to require that such a direction could be reached by the exploration policy. Existing algorithms with $O(H)$ deployment complexity (Huang et al., 2022; Qiao & Wang, 2022) avoids such an issue by assuming that any direction can be reached sufficiently frequently by some policy, in which case designing an exploration policy that can reach directions in the feature space is feasible. However, since we do not assume explorability or reachability of the underlying linear MDP as in prior work (Huang et al., 2022; Qiao & Wang, 2022), we must handle such directions carefully.

If one simply chooses to ignore such infrequent directions, the error accumulated for handling such directions would in fact blow up exponentially, rendering the final sample complexity exponential in the feature dimension $d$ or the horizon length $H$. In fact, such an issue occurs even in the simpler tabular setting. In the tabular setting, having some directions that cannot be reached is equivalent to having some state-action pairs that cannot be reached by any policy, and in order to handle such states, prior work on deployment efficient RL algorithms (Qiao et al., 2022) applied absorbing MDP to ignore those "hard to visit" states. More specifically, once the algorithm detects that some state cannot be reached by any policy, that state would be directed to a dummy state in the absorbing MDP. Since we only direct states that are hard to visit to dummy states, the error accumulated during the whole process would be additive as we have more layers, which gives a polynomial sample complexity. Indeed, this is a high-level approach of the algorithm in Qiao et al. (2022).

On the other hand, for the linear MDP setting without the reachability assumption, handling infrequent directions is much more complicated. In the tabular setting, designing exploration policies is relatively simple since we can simply plan a policy for each individual state. On the other hand, for the linear MDP setting, we need to build the exploration policy (which is a mixture of deterministic policies) in an iterative manner. Given directions that can already be reached by the current exploration policy, we need to set the reward function appropriately to encourage exploring directions that cannot be reached currently. More concretely, suppose the $\Lambda = \mathbb{E}[\phi\phi^\top]$ is the information matrix induced by the current exploration policy, for each state-action pair $(s, a)$ with feature $\phi(s, a)$, the

reward function $r(s, a)$ would be set to $\phi(s, a)^\top \Lambda^{-1} \phi(s, a)$. We then plan a new policy for the current quadratic reward function, and test whether new policy can indeed reach some new direction, both by utilizing the offline dataset. If the algorithm can no longer find any new direction that can be reached, we then proceed to the next layer. It can be shown that the total number of directions found during the whole process would be small, by using a standard potential function argument based on the determinant of the information matrix. Note that in order to test whether new policy can indeed reach some new direction, we need to estimate the information matrix $\Lambda = \mathbb{E}[\phi\phi^\top]$ of the new policy, again by utilizing the offline dataset.

Note that by assuming reachability or explorability of the feature space, we no longer need to build the exploration policy iteratively since the whole feature space can be reached and therefore one can resort to approaches based on optimal experiment design. Indeed, this is the main idea behind previous work (Qiao & Wang, 2022). However, such an approach heavily relies on reachability or explorability of the feature space, which is one of the main technical challenges we aim to tackle in this paper.

**Handling Bias Induced by Infrequent Directions.** As mentioned earlier, we heavily rely on the offline dataset obtained in previous layers for the purpose the offline policy optimization (planning for the current quadratic reward function) and offline policy evaluations (for estimating the information matrix). Moreover, since we do not assume reachability of the feature space, there are always directions that cannot be reached by the exploration policy, and therefore, it is impossible for the offline dataset to cover the whole feature space. Imperfect coverage of the offline dataset will introduce additional error for the purpose policy optimization and policy evaluation due to the bias induced by infrequent directions. Although the error accumulated during offline policy optimization can be handle relatively easily, since a global argument based comparing the groundtruth MDP and the MDP after ignoring infrequent directions would suffice, the error accumulated during offline policy evaluation is much more severe since the estimated information matrices would be used for deciding the next quadratic reward function. If not handled properly, the error will accumulate multiplicatively as we proceed to the next layer, rendering the final sample complexity exponential. Again, we note that by assuming reachability or explorability of the feature space as in prior work (Qiao & Wang, 2022), such an issue will not occur since the offline dataset is guaranteed to cover the whole feature space.

To handle such an issue, our new idea is to make sure the error of offline policy evaluation for estimating information matrices is always *multiplicative with respect to the information matrix to be evaluated*. More specifically, during the evaluation algorithm, if we encounter some state-action pair with feature $\phi = \phi(s, a)$, we would add $\phi\phi^\top$ to the evaluation result $\Lambda$ only when $\phi^\top \Lambda^{-1} \phi$ is small, to ensure a multiplicative estimation error. However, this will introduce another chicken-and-egg situation: without knowing the groundtruth information matrix $\Lambda$, it is impossible to test whether $\phi^\top \Lambda^{-1} \phi$ is small or not. To solve this issue, we use another iterative process to estimate the information matrix. Initially, we set the information matrix to be the identity matrix. In each iteration, in order to test whether $\phi^\top \Lambda^{-1} \phi$ is small or not, we use information matrix $\Lambda$ obtained during the previous iteration, adding up $\phi\phi^\top$ for those $\phi$ that passed the test to form the new information matrix, and proceed to the next iteration. We stop the whole iteration process if the two information matrices obtained in two consecutive iterations are close enough (in a multiplicative sense). By using another potential function argument based on the determinant of the information matrix, it can be shown that the iterative process stops with small number of rounds. Such an idea is another major technical contribution of the present work.

**Handling Dependency Issues by Independent Copies.** According to the discussion above, our final algorithm involves two iterative processes, and since the results of different iterations all rely on the same offline dataset, these results are subtly coupled with each other. Fortunately, such dependency issues are relatively easy to handle, since we can simply make independent copies of the offline dataset by repeatedly sampling trajectories by following exploration policy with fresh randomness.

Our final algorithm is a careful combination of all ideas mentioned above.

## 5 ALGORITHMS

In this section, we introduce our algorithms. The parameter settings are postpone to Appendix A due to space limitation.

**Main algorithm: `Sampling` (Algorithm 1).** In the main algorithm, the learner collects samples layer by layer. In each deployment, the learner assumes that it has learned enough information about previous layers, and focuses on learning the current layer. In the sub-problem of learning one layer, `Policy-Design` is called to design the policy to explore current layer given previous samples, and `Policy-Execution` is called to play this policy to collect samples.

In each call of `Policy-Design`, there are $m$ optimization sub-problems (see line 6 Algorithm 2) and $m$ off-value-evaluation sub-problems (see line 12 Algorithm 2). As mentioned in Section 4, we collect multiple copies of datasets, and use a group of new datasets to solve each sub-problem. As a result, the datasets are independent of the reward and policy in the sub-problem. More precisely, we collect $(2m^2 + 1) \cdot H$ copies for each datasets to help solve the $2mH$ sub-problems, where each dataset consists of $N$ sample points. We refer readers to Algorithm 6 for more details about how to collect samples.

**Remark 1.** *In Algorithm 1, the first layer is a slightly different from other layers because of unknown initial distribution, where the local optimal design (see Lemma 7) is used to reduce one deployment (see the algorithm and analysis in Appendix E).*

**`Policy-Design` (Algorithm 2).** We consider learning the $h$-th layer. Given datasets in the first $h-1$ layers, the learner first designs reward function with form $r_h(s, a) \leftarrow \min\left\{\phi_h^\top(s, a)\Lambda^{-1}\phi_h(s, a), 1\right\}$, where $\Lambda$ is the current information matrix. We hope to update $\Lambda$ as

$$\Lambda_{\text{new}} \leftarrow \mathbb{E}_{\pi_{\text{old}}}\left[\phi_h \phi_h^\top\right] + \Lambda_{\text{old}}, \tag{2}$$

where $\pi_{\text{old}}$ is a near-optimal policy with respect to the reward $r_{\text{old}} = \min\{\phi_h^\top \Lambda_{\text{old}}^{-1}\phi_h, 1\}$. By iteratively running this process, we can finally obtain some $\Lambda$ such that $\max_\pi \mathbb{E}_\pi\left[\min\{\phi_h^\top \Lambda^{-1}\phi_h, 1\}\right]$ is small. As discussed in Section 4, it might be improper to add $\mathbb{E}_{\pi_{\text{old}}}\left[\phi_h \phi_h^\top\right]$ to $\Lambda$ directly due to the infrequent directions. Therefore, we need to truncate the infrequent directions in the distribution $\pi_{\text{old}}$, and evaluate the truncated matrix with off-line dateset. Below we explain how to address this problem by ALgorithm 3.

**`Matrix-Evaluation` (Algorithm 3).** In this algorithm, the input is a policy $\pi$ and a group of datasets. The target is to truncate the infrequent directions under $\pi$, and evaluate the information matrix after truncating the infrequent directions. To describe the high-level idea in the algorithm, we assume $D$ is an distribution over $\mathbb{R}^d$ and consider to truncated the infrequent direction under $d$. We assume that $D$ is known to the learner. So one can immediately compute $\Lambda = \mathbb{E}_D[\phi\phi^\top]$ and compute the infrequent directions $\phi$ such that $\phi^\top \Lambda^{-1}\phi$ is large. The next step is to re-scale $\phi$ with $w(\phi) \cdot \phi$ such that $w^2\phi^\top \Lambda^{-1}\phi$ is small. However, after truncation, the new information matrix would be $\Lambda_{\text{new}} = \mathbb{E}_{\phi \sim D}[w^2(\phi)\phi\phi^\top] \preceq \Lambda$, which means that a frequent direction under $\Lambda$ might turn to be an infrequent direction under $\Lambda_{\text{new}}$. A straightforward idea is to repeat this process, until $\Lambda$ converges to some point. Let $F(\Lambda) = \mathbb{E}_{\phi \sim D}\left[\text{T}(\phi\phi^\top, c_1\Lambda)\right]$ where $c_1$ is the threshold for truncation. By iteratively running $F(\cdot)$, and noting that $F(\cdot)$ is non-increasing and the set of bounded PSD matrices is compact, the sequence $\{F^{(n)}(\Lambda)\}_{n \geq 1}$ will converge to some $\Lambda^*$ and it holds that $F(\Lambda^*) = \Lambda^*$, which means no more truncation is needed after truncation w.r.t. $\Lambda^*$. In words, the infrequent directions no longer exists. One might be worried that $\mathbf{0}$ is also a fixed point of $F(\cdot)$, so that the truncation is meaningless if $\Lambda^* = \mathbf{0}$. Fortunately, by choose $c_1$ properly large, we can show that $\Pr_{\phi \sim D}[\phi^\top(\Lambda^*)^{-1}\phi \geq c_1] = O(\epsilon)$, which means only a small portion of vectors are truncated. When $D$ is unknown, we could sample from $D$ to estimate $\mathbb{E}_D[\text{T}(\phi\phi^\top, \Lambda)]$ and play the same iteration. Incorporating this idea with the arguments of linear regression, we devise Algorithm 3 to truncate and evaluate the information matrix efficiently.

## 6 ANALYSIS

In this section, we present the formal version of the main theorem and sketch the proof.

---

**Algorithm 1** `Sampling`

---

1: $\{\mathcal{D}_0^h, \mathcal{D}_1^h\}_{h=1}^H \leftarrow$ `Ini-Sampling`;
2: **for** $h = 2, \ldots, H$ **do**
3:    $\left\{\{\pi^{i,h}\}_{i=1}^m, \check{\Lambda}_h\right\} \leftarrow$ `Policy-Design` $\left(h, \{\mathcal{D}_\tau^h(j)\}_{\tau \in [h-1], j \in [2m^2]} \cup \{\mathcal{D}_0^h(j)\}_{j \in [2m^2]}, \{\check{\Lambda}_\tau\}_{\tau \in [h-1]}, \right)$

4:    $\{\mathcal{D}_h^\tau\}_{\tau=1}^H \leftarrow$ `Policy-Execution`$(h, \{\pi^{i,h}\}_{i=1}^m, \check{\Lambda}_h)$;
5: **end for**
6: **return**: $\{\mathcal{D}_h^h(2m^2 + 1)\}_{h \in [H]}$ and $\{\check{\Lambda}_h\}_{h \in [H]}$

---

**Algorithm 2** `Policy-Design`

---

**Input:** $h$, datasets $\{\phi_{\tau,i}(j), \tilde{s}_{\tau,i}(j), \lambda_{\tau,i}(j)\}_{i \in [N], \tau \in [h-1], j \in [2m^2]}$, $\{s_{1,i}(j)\}_{i \in [N], j \in [2m]}$, block
matrix $\{\check{\Lambda}_\tau\}_{\tau \in [h-1]}$;
**Initialization:** $\Lambda_h^0 = \zeta \mathbf{I}$;
**for** $\ell = 1, 2, \ldots, m$ **do**
  $r_h^\ell(s, a) \leftarrow \min\{\phi_h(s, a)^\top (\Lambda_h^{\ell-1})^{-1} \phi_h(s, a), 1\}$ for all $(s, a)$;
  $r_\tau^\ell(s, a) \leftarrow 0$ for $\tau \neq h$ and all $(s, a)$;
  $\{\pi^\ell, v_h^\ell\} \leftarrow$ `Opt`$(h, \{\phi_{\tau,i}(m^2 + \ell), \tilde{s}_{\tau,i}(m^2 + \ell), \lambda_{\tau,i}(m^2 + \ell)\}_{i \in [N], \tau \in [h-1]}, \{s_{1,i}(m^2 + \ell)\}_{i=1}^N, r^\ell := \{r_\tau^\ell\}_{\tau \in [H]})$;
  // *Let $Y_{\tau,i}(a : b)$ denote the dataset $\{Y_{\tau,i}(j)\}_{j=a}^b$ for $a \leq b$ for $Y = \phi, \tilde{s}, \lambda$ and $s_1$;*
  $\check{\mathcal{D}} \leftarrow \{\phi_{\tau,i}((\ell-1)m-1 : \ell m), \tilde{s}_{\tau,i}((\ell-1)m-1 : \ell m), \lambda_{\tau,i}((\ell-1)m-1 : \ell m)\}_{i \in [N], \tau \in [h-1]}$;
  $\check{\mathcal{D}}_0 \leftarrow \{s_{1,i}((\ell-1)m-1 : \ell m)\}_{i=1}^N$;
  // *Feed independent sub-datasets to* `Matrix-Evaluation`;
  $\{\bar{\Lambda}_h^\ell, \check{\Lambda}_h^\ell\} \leftarrow$ `Matrix` $-$ `Evaluation`$(h, \pi^\ell, \check{\mathcal{D}}, \check{\mathcal{D}}_0)$;
  $\Lambda_h^\ell \leftarrow \Lambda_h^{\ell-1} + \bar{\Lambda}_h^\ell$;
**end for**
**return:** $\{\pi^{i,h}\}_{i=1}^m$ and $\check{\Lambda}_h \leftarrow \Lambda_h^m$.

---

**Theorem 4.** *By running Algorithm 1, the learner can collect a group of trajectories such that: with probability $1 - \delta$, for any reward kernel $\{\theta_h\}_{h \in [H]}$ satisfying Assumption 2, the learner can return an $\epsilon$-optimal policy $\pi$. That is,*

$$\mathbb{E}_\pi \left[ \sum_{h=1}^H \phi_h^\top(s_h, a_h)\theta_h \right] \geq \max_{\pi'} \mathbb{E}_{\pi'} \left[ \sum_{h=1}^H \phi_h^\top(s_h, a_h)\theta_h \right] - \epsilon.$$

*Moreover, Algorithm 1 consists of $H$ deployments, where the number of episodes in each deployment is $\tilde{O}\left(\frac{d^9 H^{14}}{\epsilon^5}\right)$[2].*

*Proof.* We first count the number of deployments. There is one deployment in line 1. For each $h = 2, \ldots, H$, there is one deployment in line 4. We then conclude that the number of deployments is $H$. On the other hand, the number of trajectories in each deployment is $O(H(2m + 1)N) = \tilde{O}\left(\frac{d^9 H^{14}}{\epsilon^5}\right)$.

The proof is finished by the following lemma, which proves the optimality of the learned policy. The proof is presented in Appendix C.6

**Lemma 5.** *With probability $1 - \delta$, for any reward kernel $\theta \in \{\theta_h\}_{h=1}^H$ satisfying Assumption 2, `Planning`$\left(\theta, \{\phi_{h,i}, \tilde{s}_{h,i}, \lambda_{h,i}\}_{i=1}^N\}_{h \in [H]}, \{\check{\Lambda}_h\}_{h \in [H]}\right)$ returns an $\epsilon$-optimal policy, where $\{\phi_{h,i}, \tilde{s}_{h,i}, \lambda_{h,i}\}_{i=1}^N\}_{h \in [H]}$ and $\{\check{\Lambda}_h\}_{h \in [H]}$ is the output of Algorithm 1.*

$\square$

---

[2]We use $\tilde{O}(\cdot)$ to omit logarithmic factors of $(d, H, \frac{1}{\epsilon}, \frac{1}{\delta})$.

---

**Algorithm 3** `Matrix-Evaluation`

---

1: **Input:** horizon $h$, policy $\pi$, dataset $\{\phi_{\tau,i}(j), \tilde{s}_{\tau,i}(j), \lambda_{\tau,i}(j)\}_{\tau \in [h-1], i \in [N], j \in [m]} \cup \{s_{1,i}(j)\}_{i \in [N], j \in [m]}.$

2: $\Lambda \leftarrow \mathbf{I}$;

3: **for** $j = 1, 2, \ldots, m$ **do**

4: $\quad \hat{F}_0 \leftarrow$ `T-M-Evaluation`$(\Lambda, \{\phi_{\tau,i}(j), \tilde{s}_{\tau,i}(j), \lambda_{\tau,i}(j)\}_{\tau \in [h-1], i \in [N]}, \{s_{1,i}(j)\}_{i \in [N]})$;

5: $\quad$ **if** $\hat{F}_0 + \frac{\zeta}{2x}\mathbf{I} \succeq \frac{1}{2}\Lambda$ **then**

6: $\qquad$ **break** and **return** $\left\{\hat{F}_0 + \frac{\zeta}{2x}\mathbf{I}, \Lambda\right\}$;

7: $\quad$ **else**

8: $\qquad \Lambda \leftarrow \hat{F}_0$;

9: $\quad$ **end if**

10: **end for**

11: **Function** : `T-M-Evaluation` $(\Lambda, \{\phi_{\tau,i}, \tilde{s}_{\tau,i}, \lambda_{\tau,i}\}_{\tau \in [h-1], i \in [N]}, \{s_{1,i}\}_{i \in [N]})$

12: $\hat{F}_h(s) \leftarrow$ `T`$(\phi_h(s, \pi_h(s))\phi_h^\top(s, \pi_h(s)), f_1\Lambda)$ for $s \in \{\tilde{s}_{h-1,i}\}_{i \geq 1}$;

13: **for** $\tau = h - 1, h - 2, \ldots, \ldots, 1$ **do**

14: $\quad X_\tau \leftarrow \sum_{i=1}^N \lambda_{\tau,i}^2 \phi_{\tau,i} \phi_{\tau,i}^\top + z\mathbf{I}$;

15: $\quad$ **for** $s \in \{\tilde{s}_{\tau-1,i}\}_{i \in [N]}$ **do**

16: $\qquad \phi \leftarrow \phi_\tau(s, \pi_\tau(s))$;

17: $\qquad$ **if** $\phi^\top \check{\Lambda}_\tau \phi \geq 1$ **then**

18: $\qquad\quad \hat{F}_\tau(s) \leftarrow \mathbf{0}$;

19: $\qquad$ **else**

20: $\qquad\quad \hat{F}_\tau(s) \leftarrow \phi^\top X_\tau^{-1} \sum_{i=1}^N \lambda_{\tau,i}^2 \phi_{\tau,i} \hat{F}_{\tau+1}(\tilde{s}_{\tau,i}) + 2x\Lambda + 2\zeta\mathbf{I}$;

21: $\qquad$ **end if**

22: $\quad$ **end for**

23: **end for**

24: **return** : $\hat{F}_0 := \frac{1}{N}\sum_{i=1}^N \hat{F}_1(s_{1,i}) + 2x\Lambda + 2\zeta\mathbf{I}$;

---

To prove Lemma 5, a central lemma is introduced as follows, which states that the output dataset of Algorithm 1 could efficiently cover all policies.

**Lemma 6.** *Let $\{\{\pi^{i,h}\}_{i=1}^m, \check{\Lambda}_h\}$ be the output of* `Policy-Design` *in Line 3 in the $h$-th iteration. With probability $1 - \frac{\delta}{2} - \frac{\delta}{2H}$, for any dataset of Algorithm 1 for the $h$-th layer $\{\phi_{h,i}, \tilde{s}_{h,i}, \lambda_{h,i}\}_{i \in [N]}$, it holds that*

    *i.* $\max_\pi \Pr_\pi\left[\phi_h^\top \check{\Lambda}_h \phi_h > 1, \phi_\tau^\top \check{\Lambda}_\tau \phi_\tau \leq 1, \forall \tau \in [h-1]\right] \leq \frac{\epsilon}{8H^2}$ *for all $h \in [H]$;*

    *ii.* $\sum_{i=1}^N \lambda_{\tau,i}^2 \phi_{\tau,i} \phi_{\tau,i}^\top + z\mathbf{I} \succeq \frac{N}{8m}\check{\Lambda}_h$ *for all $h \in [H]$;*

    *iii.* $\lambda_{h,i}^2 \phi_{h,i}^\top \check{\Lambda}_h^{-1} \phi_{h,i} \leq f_1$ *for all $h \in [H]$ and $i \in [N]$.*

The rest part of this section is devoted to sketching the proof of Lemma 6. We will prove by induction over the layer. We now assume the three conditions in Lemma 6 holds for the first $h - 1$ layers.

**Truncated MDP.** We define the truncated MDP $M_{h-1}$ by redirection all $\phi_\tau(s, a)$ to a dumb state if $\phi_\tau(s, a)^\top \check{\Lambda}_\tau^{-1} \phi_\tau(s, a) > 1$ for $\tau \in [h-1]$. More precisely, a trajectory $\{(s_\tau, a_\tau)\}_{\tau=1}^H$ under the original MDP $M$ is mapped to $\{(s_1, a_1), \ldots, (s_k, a_k), \mathbf{z}, \ldots, \mathbf{z}\}$ under $M_{h-1}$. Here $k \leq h - 1$ is the smallest integer such that $\phi_k^\top(s_k, a_k)\check{\Lambda}_k^{-1}\phi_k(s_k, a_k) > 1$ and $\mathbf{z}$ is the dumb state. If $\phi_k^\top(s_k, a_k)\check{\Lambda}_k^{-1}\phi_k(s_k, a_k) \leq 1$ for all $k \in [h-1]$, the trajectory is invariant after the truncation.

In the following analysis, we re-define $\mathbb{E}[\cdot]$ and $\Pr[\cdot]$ to be respectively the expectation and probability under $M_{h-1}$.

*Proof of Lemma 6.* The proof for the first layer is a slightly different due to unknown initial distribution. We postpone the algorithm and proofs to Appendix E.

Below we sketch the proof for the $h$-th layer ($h \geq 2$). The missing lemmas and proofs are presented in Appendix C. We verify the three conditions as below.

**Condition (i).** By Lemma 12, with probability $1 - \frac{\delta}{8H}$,

$$\max_\pi \mathbb{E}_\pi \left[ \min\{\phi_h^\top \check{\Lambda}_h^{-1} \phi_h, 1\} \right] \leq \frac{\epsilon}{8H^2}, \tag{3}$$

which implies that

$$\max_\pi \Pr_\pi \left[ \phi_h^\top \check{\Lambda}_h^{-1} \phi_h > 1 \right] \leq \frac{\epsilon}{8H^2}. \tag{4}$$

The proof is finished by noting equation 4 under the truncated MDP $M_{h-1}$ is equivalent to (i).

**Condition (ii).** By Lemma 16 , with probability $1 - \frac{\delta}{16H}$, it holds that

$$\sum_{i=1}^N \lambda_{h,i}^2 \phi_{h,i} \phi_{h,i}^\top + z\mathbf{I} \succeq \frac{N}{8m} \check{\Lambda}_h$$

for all sub-datasets $\{\phi_{h,i}, \tilde{s}_{h,i}, \lambda_{h,i}\}_{i=1}^N$.

**Condition (iii).** To verify the third condition, it suffices to note the definition that : $\lambda_{h,j} = \min \left\{ \sqrt{\frac{f_1}{\phi_{h,i}^\top \check{\Lambda}_h^{-1} \phi_{h,j}}}, 1 \right\}$ (See Algorithm 6).

$\square$

## 7 CONCLUSION

In this work, we design a new RL algorithm whose sample complexity is polynomial in the feature dimension and horizon length, while achieving nearly optimal deployment complexity for linear MDPs. Moreover, our algorithm works under the reward-free exploration setting, and does not require any additional assumptions on the underlying MDP. In our new algorithm and analysis, we propose new methods to truncate state-action pairs in a data-dependent manner, and design efficient offline algorithms for evaluating information matrices. Given our new results, an interesting future direction is to generalize our new techniques to other RL problems. For example, for function classes with bounded eluder dimension (Wang et al., 2020b; Kong et al., 2021; Zhao et al., 2023) , it would be interesting to design RL algorithm with nearly optimal $O(H)$ deployment complexity and polynomial sample complexity without relying on any additional assumptions.

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

## A  PARAMETER SETTINGS AND NOTATIONS

Assume $d, H \geq 40$, $\epsilon \leq \frac{1}{40}$. Set $x = \frac{1}{100dH}$, $f_1 = \frac{320dH^2}{\epsilon}$, $\zeta = \frac{\epsilon^5}{10000d^5H^{15}}$, $\xi = \left(\frac{\epsilon}{10d^2H^2}\right)^{10}$, $z = \frac{100000\epsilon^2}{d^2H^5}$, $m = \frac{32000d^2H^3}{\epsilon}$, $N = \frac{10^9 d^5 H^7 \log\left(\frac{dH}{\epsilon\delta}\right)}{\epsilon^3}$. Total number of samples $H^2(2m^2+1)N = \tilde{O}\left(\frac{d^9H^{15}}{\epsilon^5}\right)$. The number of trajectories in each deployment is $\tilde{O}\left(\frac{d^9H^{14}}{\epsilon^5}\right)$.

We also present a table of notations as follows.

Table 1: Additional Notations.

| Notation | Comments |
|---|---|
| $P_h(\cdot|s,a)$ | the transition probability for the triple $(h,s,a)$ |
| $r_h(s,a)$ | the reward expectation for the triple $(h,s,a)$ |
| $\phi_h(s,a)$ | the $d$-dimensional feature vector for the triple $(h,s,a)$ |
| $\mu_h$ | the probability transition kernel be such that $P_h(\cdot|,s,a) = \mu\phi_h(s,a)$ |
| $\theta_h(v)$ | the $d$-dimensional payoff vector defined as $\mu_h^\top v$ |
| $\mathtt{T}(\cdot,\cdot)$ | the truncation function |
| $N$ | the number of datapoints in one dataset |
| $\{\phi_\tau, \tilde{s}_\tau, \lambda\}$ | one datapoint from the $\tau$-th layer, $\phi_\tau$: the feature vector, $\tilde{s}_\tau$: the next state, $\lambda$: weight |
| $\{\phi_{\tau,i}, \tilde{s}_{\tau,i}, \lambda_{\tau,i}\}_{i=1}^N$ | an independent dataset from the $\tau$-th layer |
| $\zeta$ | the regularization parameter |
| $\xi$ | the discretization parameter |
| $\mathcal{E}_1(\phi, v)$ | the concentration event for feature $\phi$ and value $v$ w.r.t. an independent dataset |
| $\mathcal{E}_2(\phi, f)$ | the concentration event for feature $\phi$ and matrix value $f$ w.r.t. an independent dataset |

# B  TECHNICAL LEMMAS

**Lemma 7** (General Equivalence Theorem in Kiefer & Wolfowitz (1960)). *For any bounded subset $X \subset \mathbb{R}^d$, there exists a distribution $\mathcal{K}(X)$ supported on $X$, such that for any $\epsilon > 0$, it holds that*

$$\max_{x \in X} x^\top \left(\epsilon \mathbf{I} + \mathbb{E}_{y \sim \mathcal{K}(X)}[yy^\top]\right)^{-1} x \le d. \tag{5}$$

*Furthermore, there exists a mapping $\pi^{\mathtt{G}}$, which maps a context $X$ to a distribution over $X$ such that*

$$\max_{x \in X} x^\top (\epsilon \mathbf{I} + \mathbb{E}_{y \sim \pi^{\mathtt{G}}(X)}[yy^\top])^{-1} x \le 2d. \tag{6}$$

*In particular, when $\mathrm{supp}(X)$ has a finite size, $\pi^{\mathtt{G}}(X)$ could be implemented within $\mathrm{poly}(|\mathrm{supp}(X)|, \log(1/\epsilon))$ time.*

**Lemma 8.** *Assume $0 \le \kappa \le 0.1$. Let $\Lambda^0 = \zeta \mathbf{I}$. For each $i \ge 1$, let $D^i$ be a distribution over $\mathbb{R}^d$ satisfying that*

$$\mathbb{E}_{\phi \sim D^i}\left[\min\left\{\mathrm{Trace}\left(\phi\phi^\top(\Lambda^{i-1})^{-1}\right), 1\right\}\right] \ge \kappa \tag{7}$$

*and*

$$\Lambda^i \succeq \Lambda^{i-1} + \mathbb{E}_{\phi \sim D^i}[\phi\phi^\top].$$

*Then we have that*

$$\log(\det(\Lambda^n)) - \log(\det(\Lambda^0)) \ge \frac{n\kappa}{4} \tag{8}$$

*for any $n \ge 1$.*

*Proof.* Fix $i \ge 1$. Note that equation 7 is equivalent to

$$\mathbb{E}_{\phi \sim D^i}\left[\min\{\phi^\top(\Lambda^{i-1})^{-1}\phi, 1\}\right] \ge z. \tag{9}$$

Let $W := \mathbb{E}_{\phi \sim D^i}\left[\mathtt{T}(\phi\phi^\top, \Lambda^{i-1})\right] \preceq \mathbb{E}_{\phi \sim D^i}\left[\phi\phi^\top\right]$. By definition, it holds that $W \preceq \Lambda^{i-1}$ and $W + \Lambda^{i-1} \preceq 2\Lambda^{i-1}$. We then have that

$$\log(\det(\Lambda^i)) - \log(\det(\Lambda^{i-1})) \ge \log(\det(\Lambda^{i-1} + W)) - \log(\det(\Lambda^{i-1}))$$

$$= \log\left(\det(\mathbf{I} + (\Lambda^{i-1})^{-1/2}W(\Lambda^{i-1})^{-1/2})\right)$$

$$= \log\left(\det\left(\mathbf{I} + (\Lambda^{i-1})^{-1/2}\mathbb{E}_{\phi \sim D}\left[\mathtt{T}(\phi\phi^\top, \Lambda^{i-1})\right](\Lambda^{i-1})^{-1/2}\right)\right)$$

$$\ge \frac{1}{4}\mathbb{E}_{\phi \sim D^i}\left[\mathrm{Trace}(\mathtt{T}(\phi\phi^\top, \Lambda^{i-1})(\Lambda^{i-1})^{-1})\right]$$

$$\ge \frac{\kappa}{4}. \tag{10}$$

The proof is completed by taking sum over $i$ from 1 to $n$.

$\square$

### B.1 CONCENTRATION INEQUALITIES

**Lemma 9.** *Let $X_1, X_2, ..., X_n$ be a group of zero-mean matrices such that $-\Lambda \preceq X_i \preceq \Lambda$ with probability 1 for all $i \in [N]$. Let $w_1, w_2, ..., w_n$ be a group of reals. With probability $1 - \delta$,*

$$-2\sqrt{\sum_{i=1}^{n} w_i^2 \log(2d/\delta)\Lambda} - 2\max_i |w_i| \log(2d/\delta)\Lambda \preceq \sum_{i=1}^{n} w_i X_i$$

$$\preceq 2\sqrt{\sum_{i=1}^{n} w_i^2 \log(2d/\delta)\Lambda} + 2\max_i |w_i| \log(2d/\delta)\Lambda. \quad (11)$$

*Proof.* Without loss of generality, we assume $\Lambda = \mathbf{I}$. For $0 \le t \le \frac{1}{\max_i |w_i|}$, define

$$E_k = \mathbb{E}\left[\text{Trace}\left(\exp\left(t\sum_{i=1}^{k} w_i X_i - 2t^2 \sum_{i=1}^{k} w_i^2 \mathbf{I}\right)\right)\right].$$

Then we have that

$$\mathbb{E}[E_k | X_{1:k-1}] \le \mathbb{E}\left[\text{Trace}\left(\exp\left(\log\left(\mathbb{E}[\exp(tw_k X_k)|X_{1:k-1}]\right) + t\sum_{i=1}^{k-1} w_i X_i\right)\right)\right]$$

$$\mathbb{E}\left[\text{Trace}\left(\exp\left(\log(\mathbb{E}[\exp(tw_k X_k)|X_{1:k-1}])\right) - 2t^2 w_k^2 \mathbf{I} + t\sum_{i=1}^{k-1} w_i X_i - 2t^2 \sum_{i=1}^{k-1} w_i^2 \mathbf{I}\right)\right] \quad (12)$$

$$\le \mathbb{E}\left[\text{Trace}\left(t\sum_{i=1}^{k-1} w_i X_i - 2t^2 \sum_{i=1}^{k-1} w_i^2 \mathbf{I}\right)\right]$$

$$= E_{k-1},$$

where the first inequality is by Lieb's inequality (see Theorem 3.2, Tropp (2012)) and the second inequality is by the fact that $\mathbb{E}[\exp(tw_k X_k)] \preceq \exp(2t^2 w_k^2)\mathbf{I}$. As a result, we learn that $\mathbb{E}[E_n] \le \mathbb{E}[E_0] = d$, which means that with probability $1 - \delta/2$, the maximal eigenvalue of $\sum_{i=1}^{k} w_i X_i$ is at most $2\sqrt{\sum_{i=1}^{n} w_i^2 \log(2d/\delta)} + 2\max_i |w_i| \log(2d/\delta)$. Similar arguments work for the other side. The proof is completed. $\qquad\square$

## C MISSING LEMMAS AND PROOFS

### C.1 STATEMENT AND PROOF OF LEMMA 10

**Lemma 10.** *Recall $x = \frac{1}{100Hd} \ge 60\sqrt{\frac{md\log\left(\frac{dH}{\epsilon\delta}\right)}{N}}$. Define $F_h(s) := \hat{F}_h(s) = \mathtt{T}(\phi_h(s, \pi_h(s))\phi_h^\top(s, \pi_h(s)), f_1\Lambda)$. For $\tau = h - 1, h - 2, \ldots, 1$, we define $F_\tau(s) = \mathbb{E}_{s' \sim P_{\tau,s,\pi_\tau(s)}}[F_{\tau+1}(s') \cdot \mathbb{I}[\phi^\top \mathring{\Lambda}_\tau^{-1}\phi \le 1]]$ and $F_0 = \mathbb{E}_{s_1 \sim d_{\text{ini}}}[F_1(s_1)]$.*

*Let $\hat{F}_0$ be the output of the subroutine $\mathtt{T\text{-}M\text{-}Evaluation}$ in Algorithm 3 with input $\Lambda$. we have that*

$$(1 - 3Hx)F_0 \preceq \hat{F}_0 \preceq (1 + 3Hx)F_0 + 4Hx\Lambda + 4H\zeta\mathbf{I}.$$

*Proof.* It is obvious that $F_\tau(s)$ is PSD for any proper $\tau$ and $s$. Let $\tilde{s}_{0,i} = s_{1,i}$ for $1 \le i \le N$. We prove by induction that

$$(1 - 3(h - \tau)x)F_\tau(s) \preceq \hat{F}_\tau(s) \preceq (1 + 3(h - \tau)x)F_\tau(s) + 4(h - \tau)x\Lambda + 4(h - \tau)\zeta\mathbf{I} \quad (13)$$

for any $1 \le \tau \le h$ and $s \in \{\tilde{s}_{\tau-1,i}\}_{i \ge 1}$.

For $\tau = h$, we have that $\hat{F}_\tau(s) = F_\tau(s)$ for any $s \in \mathcal{S}$. Now we assume that equation 13 holds for $\tau = \ell \geq 2$. Recall that $X_\tau = \sum_{i=1}^N \lambda_{\ell-1,i}^2 \phi_{\ell-1,i} \phi_{\ell-1,i}^\top + z\mathbf{I}$. By definition, we have that for $s \in \{\tilde{s}_{\ell-2,i}\}_{i \geq 1}$

$$
\hat{F}_{\ell-1}(s) = \phi_{\ell-1}(s, \pi_{\ell-1}(s))^\top X_\tau^{-1} \sum_{i=1}^N \lambda_{\ell-1,i}^2 \phi_{\ell-1,i} \hat{F}_\ell(\tilde{s}_{\ell-1,i}) + 2x\Lambda + 2\zeta\mathbf{I}
$$

$$
= \mathbb{E}_{s' \sim P_{\ell-1,s,\pi_{\ell-1}(s)}} \left[ \hat{F}_\ell(s') \right] + \Delta_{\ell-1}^{(1)}(s) + 2x\Lambda + 2\zeta\mathbf{I}
$$

$$
= \mathbb{E}_{s' \sim P_{\ell-1,s,\pi_{\ell-1}(s)}} [F_\ell(s)] + \Delta_{\ell-1}^{(1)}(s) + \Delta_{\ell-1}^{(2)}(s) + 2x\Lambda + 2\zeta\mathbf{I}
$$

$$
= F_{\ell-1}(s) + \Delta_{\ell-1}^{(1)}(s) + \Delta_{\ell-1}^{(2)}(s) + 2x\Lambda + 2\zeta\mathbf{I}, \tag{14}
$$

where

$$
\Delta_{\ell-1}^{(1)}(s) = \phi_{\ell-1}(s, \pi_{\ell-1}(s))^\top X_\tau^{-1} \sum_{i=1}^N \lambda_{\ell-1,i}^2 \phi_{\ell-1,i} \hat{F}_\ell(\tilde{s}_{\ell-1,i}) - \mathbb{E}_{s' \sim P_{\ell-1,s\pi_{\ell-1}(s)}} \left[ \hat{F}_\ell(s') \right]
$$

$$
= \phi_{\ell-1}(s, \pi_{\ell-1}(s))^\top X_\tau^{-1} \sum_{i=1}^N \lambda_{\ell-1,i}^2 \phi_{\ell-1,i} \hat{F}_\ell(\tilde{s}_{\ell-1,i}) - \phi_{\ell-1}(s, \pi_{\ell-1}(s))^\top \mu_{\ell-1}^\top \hat{F}_\ell(\cdot); \tag{15}
$$

$$
\Delta_{\ell-1}^{(2)}(s) = \mathbb{E}_{s' \sim P_{\ell-1,s,\pi_{\ell-1}(s)}} \left[ \hat{F}_\ell(s) - F_\ell(s) \right]. \tag{16}
$$

By the induction assumption, we have that

$$
0 \preceq (1 - 3(h-\ell)x) F_\ell(s) \preceq \hat{F}_\ell(s) \preceq (1 + 3(h-\tau)x) F_\ell(x) + 4(h-\tau)x\Lambda + 4(h-\tau)\zeta\mathbf{I} \preceq 2\Lambda + 4h\zeta\mathbf{I}.
$$

By Lemma 11, with probability $1 - \frac{\delta}{16mH^2}$ it holds that

$$
\Delta_{\ell-1}^{(1)}(s) \preceq 2x\Lambda + (3hx+1)\zeta\mathbf{I} \preceq 2x\Lambda + 2\zeta\mathbf{I}; \tag{17}
$$

$$
\Delta_{\ell-1}^{(1)}(s) \succeq -2x\Lambda - (3hx+1)\zeta\mathbf{I} \succeq -2x\Lambda - 2\zeta\mathbf{I}. \tag{18}
$$

For the second term $\Delta_{\ell-1}^{(2)}(s)$, by the induction condition, we have that

$$
\Delta_{\ell-1}^{(2)}(s) \preceq 3(h-\ell)x\mathbb{E}_{s' \sim P_{\ell-1,s,\pi_{\ell-1}(s)}} [F_\ell(s')] + 4(h-\ell)x\Lambda + 4(h-\ell)\zeta\mathbf{I}
$$

$$
= 3(h-\ell)x F_{\ell-1}(s) + 4(h-\ell)x\Lambda + 4(h-\ell)\zeta\mathbf{I}; \tag{19}
$$

$$
\Delta_{\ell-1}^{(2)}(s) \succeq -3(h-\ell)x\mathbb{E}_{s' \sim P_{\ell-1,s,\pi_{\ell-1}(s)}} [F_\ell(s')]
$$

$$
= -3(h-\ell)x F_{\ell-1}(s). \tag{20}
$$

Putting all together and noting that $x \leq \frac{1}{100dH}$, we learn that

$$
\hat{F}_{\ell-1}(s) - F_{\ell-1}(s) = \Delta_{\ell-1}^{(1)}(s) + \Delta_{\ell-1}^{(2)}(s) + 2x\Lambda + 2\zeta\mathbf{I}
$$

$$
\preceq 2x\Lambda + 2\zeta\mathbf{I} + (3(h-\ell)x F_{\ell-1}(s) + 4(h-\ell)x\Lambda + 4(h-\ell)\zeta\mathbf{I})
$$

$$
\preceq 3(h-\ell+1)x F_{\ell-1}(s) + 4(h-\ell+1)x\Lambda + 4(h-\ell+1)\zeta\mathbf{I}; \tag{21}
$$

$$
\hat{F}_{\ell-1}(s) - F_{\ell-1}(s) = \Delta_{\ell-1}^{(1)}(s) + \Delta_{\ell-1}^{(2)}(s) + 2x\Lambda + 2\zeta\mathbf{I}
$$

$$
\succeq -x\Lambda - \zeta\mathbf{I} - 3(h-\ell)x F_{\ell-1}(s) + 2x\Lambda + 2\zeta\mathbf{I}
$$

$$
\succeq -3(h-\ell+1)x F_{\ell-1}(s); \tag{22}
$$

The proof of equation 13 is finished.

Note that

$$
\hat{F}_0 - F_0 = \hat{F}_0 - \mathbb{E}_{s_1 \sim d_{ini}}[\hat{F}_1(s_1)] + \mathbb{E}_{s_1 \sim d_{ini}}[\hat{F}_1(s_1) - F_1(s_1)]
$$

$$
= \frac{1}{N} \sum_{i=1}^N \hat{F}_1(s_{1,i}) - \mathbb{E}_{s_1 \sim d_{ini}}[\hat{F}_1(s_1)] + \mathbb{E}_{s_1 \sim d_{ini}}[\hat{F}_1(s_1) - F_1(s_1)] + 2x\Lambda + 2\zeta\mathbf{I}.
$$

Using the induction condition, we have that

$$0 \preceq (1-3(H-1))F_1(s) \preceq \hat{F}_1(s) \preceq (1+3(H-1)x)F_1(s)+4(H-1)x\Lambda+4(H-1)\zeta\mathbf{I} \preceq 2\Lambda+4H\zeta\mathbf{I}.$$

By Lemma 9, with probability $1 - \delta$,

$$\frac{1}{N}\sum_{i=1}^{N}\hat{F}_1(s_{1,i}) - \mathbb{E}_{s_1\sim d_{ini}}[\hat{F}_1(s_1)] \preceq 2x\Lambda + 2\zeta\mathbf{I};$$

$$\frac{1}{N}\sum_{i=1}^{N}\hat{F}_1(s_{1,i}) - \mathbb{E}_{s_1\sim d_{ini}}[\hat{F}_1(s_1)] \succeq -2x\Lambda - 2\zeta\mathbf{I}.$$

Based on equation 13, we have that

$$\mathbb{E}_{s_1\sim d_{ini}}[\hat{F}_1(s_1) - F_1(s_1)] \preceq 3(h-1)xF_0 + 3(h-1)x\Lambda + 3(h-1)\zeta\mathbf{I}$$
$$= 3(h-1)x\mathbb{E}_{s_1\sim d_{ini}}[F_1(s_1)] + 4(h-1)x\Lambda + 4(h-1)\zeta\mathbf{I};$$
$$\mathbb{E}_{s_1\sim d_{ini}}[\hat{F}_1(s_1) - F_1(s_1)] \succeq -3(h-1)x\mathbb{E}_{s_1\sim d_{ini}}[F_1(s_1)]$$
$$= -3(h-1)xF_0.$$

As a result, we obtain that

$$(1 - 3hx)F_0 \preceq \hat{F}_0 \preceq (1 + 3hx)F_0 + 4hx\Lambda + 4h\zeta\mathbf{I}.$$

The proof is finished.

$\square$

### C.2 STATEMENT AND PROOF OF LEMMA 11

**Lemma 11.** *Fix $f : \mathcal{S} \to \mathbb{R}^{d^2}$ such that $0 \preceq f(s) \preceq \Lambda, \forall s \in \mathcal{S}$ for some PSD matrix $\Lambda$. Let $\{\phi_{\tau,i}, \tilde{s}_{\tau,i}, \lambda_{\tau,i}\}_{i=1}^{N}$ be a dataset from the $\tau$-th layer. Assume $\{\phi_{\tau,i}, \tilde{s}_{\tau,i}, \lambda_{\tau,i}\}_{i=1}^{N}$ is independent of $f$. Let $X_\tau = \sum_{i=1}^{N} \lambda_{\tau,i}^2 \phi_{\tau,i}\phi_{\tau,i}^\top + z\mathbf{I}$. Then with probability $1 - \frac{\delta}{16mH^2}$*

$$\left| \phi^\top \mu_\tau^\top f - \phi^\top X_\tau^{-1} \sum_{i=1}^{N} \lambda_{\tau,i}^2 \phi_{\tau,i} f(\tilde{s}_{\tau,i}) \right| \preceq 60\sqrt{\frac{md\log\left(\frac{dH}{\epsilon\delta}\right)}{N}} \cdot \Lambda \qquad (23)$$

*holds for any $\phi \in \mathbb{R}^2$ such that $\|\phi\|_2 \le 1$ and $\phi^\top \check{\Lambda}_\tau^{-1}\phi \le 1$.*

*Proof.* By the induction assumption (i) and (iii), we have that $X_\tau \succeq \frac{N}{8m}\check{\Lambda}_\tau$ for $1 \le \tau \le h - 1$ and $\max_i \phi_{\tau,i}^\top X_\tau^{-1}\phi \le f_1$. By Lemma 14, with probability $1 - \frac{\delta}{16mH^2}$, we have that

$$\left| \phi^\top \mu_\tau^\top f - \phi^\top X_\tau^{-1} \sum_{i=1}^{N} \lambda_{\tau,i}^2 \phi_{\tau,i} f(\tilde{s}_{\tau,i}) \right|$$

$$\preceq \left( 16\sqrt{\phi^\top X_\tau^{-1}\phi d\log(\frac{dH}{\epsilon\delta})} + 8\sqrt{\max_i \phi_{\tau,i}^\top X_\tau^{-1}\phi_{\tau,i}\phi^\top X_\tau^{-1}\phi} \cdot d\log\left(\frac{dH}{\epsilon\delta}\right) + \zeta \right)\Lambda$$

$$\preceq 60\sqrt{\frac{md\log\left(\frac{dH}{\epsilon\delta}\right)}{N}} \cdot \Lambda.$$

$\square$

## C.3   STATEMENT AND PROOF OF LEMMA 12

**Lemma 12.** *Recall the definition of $\check{\Lambda}_h = \Lambda_h^m$ in Algorithm 1. With probability $1 - \frac{\delta}{8H}$, it holds that*

$$\max_\pi \mathbb{E}_\pi \left[ \min\{\phi_h^\top \check{\Lambda}_h^{-1} \phi_h, 1\} \right] \leq \max \left\{ \frac{40 d \log(3m/\zeta)}{m}, \frac{4}{3} B + \frac{2d}{f_1} \right\} \leq \frac{\epsilon}{8H^2}.$$

*Proof.* Recall the definition of $\{\Lambda_h^\ell\}_{\ell=1}^m$, $\{\bar{\Lambda}_h^\ell\}_{\ell=1}^m$ and $\{\check{\Lambda}_h^\ell\}_{\ell=1}^m$ in Algorithm 2. Let $y^\ell = \max_\pi \mathbb{E}_\pi \left[ \min \left\{ \phi_h^\top (\Lambda_h^\ell)^{-1} \phi_h, 1 \right\} \right]$. Then $y^\ell$ is non-increasing in $\ell$ because $\Lambda_h^\ell$ is non-decreasing in $\ell$. Let $y = y^m = \max_\pi \mathbb{E}_{M_h,\pi} \left[ \min\{\phi_h^\top \check{\Lambda}_h^{-1} \phi_h, 1\} \right]$. By Lemma 13 and Lemma 15, with probability $1 - \frac{\delta}{8mH} \cdot m = 1 - \frac{\delta}{8H}$,

$$\mathbb{E}_{\pi^\ell} \left[ \min \left\{ \text{Trace} \left( \phi_h \phi_h^\top (\Lambda_h^{\ell-1})^{-1} \right), 1 \right\} \right]$$
$$\geq \mathbb{E}_{\pi^\ell} \left[ \min\{\text{Trace}(\phi_h \phi_h^\top (\Lambda_h^{\ell-1})^{-1}), 1\} \right] - \Pr_{\pi^\ell} \left[ \phi_h^\top (\check{\Lambda}_h^{\ell-1})^{-1} \phi_h > f_1 \right]$$
$$\geq \mathbb{E}_{\pi^\ell} \left[ \min\{\text{Trace}(\phi_h \phi_h^\top (\Lambda_h^{\ell-1})^{-1}), 1\} \right] - \frac{d}{f_1(1 - 3Hx)}$$
$$\geq y^{\ell-1} - B - \frac{d}{f_1(1 - 3Hx)}$$
$$\geq y - B - \frac{d}{f_1(1 - 3Hx)}.$$

**Case i:** $y - B - \frac{d}{f_1(1-3Hx)} \geq \frac{y}{4}$. By Lemma 8 with the $D_\ell$ as the distribution of $\phi_h \cdot \min\left\{ \sqrt{\frac{f_1}{\phi_h^\top (\Lambda_h^{\ell-1})^{-1} \phi_h}}, 1 \right\}$ under $\pi^\ell$ and $\kappa = \frac{y}{10} \leq 0.1$, we have that

$$\log(\det(\Lambda_h^m)) - \log(\det(\Lambda_h^0)) \geq \frac{my}{40}. \tag{24}$$

Recall the definition of $\{\bar{\Lambda}_h^\ell\}_{\ell=1}^m$ in Algorithm 2. Using Lemma 10, we have that $\bar{\Lambda}_h^\ell \preceq 3\mathbf{I}$ and thus $\log(\det(\Lambda_h^m)) \leq d \log(3m)$. On the other hand, we have that $\log(\det(\Lambda_h^0)) = d \log(\zeta)$, which means that $\frac{my}{40} \leq d \log(3m/\zeta)$. Therefore, we have that $y \leq \frac{40 d \log(3m/\zeta)}{m} \leq \frac{\epsilon}{8H^2}$.

**Case ii:** $y - B - \frac{d}{f_1(1-3Hx)} < \frac{y}{4}$.   In this case, we have that $y \leq \frac{4}{3} B + \frac{2d}{f_1} \leq \frac{\epsilon}{8H^2}$.

$\square$

## C.4   STATEMENT AND PROOF OF LEMMA 13

**Lemma 13.** *Let $B = 2\sqrt{\frac{H^2 \log(1/\delta)}{N}} + 2\frac{H \log(1/\delta)}{N} + 2H \left( 32 \sqrt{\frac{md \log\left(\frac{dH}{\epsilon\delta}\right)}{N}} + \frac{32 md \sqrt{f_1} \log\left(\frac{dH}{\epsilon\delta}\right)}{N} \right)$.*
*Let $\{V_0^i, \pi^i\}$ be the output of $\mathtt{Opt}$ with input reward as $r^i$. With probability $1 - \frac{\delta}{8mH}$*

$$\max_\pi \mathbb{E}_\pi \left[ r_h^i(s_h) \right] - \mathbb{E}_{\pi^i} \left[ r_h^i(s_h) \right] \leq B.$$

*Proof.* Assume $w \in \mathbb{R}^{\mathcal{S}}$ satisfying $\|w\|_\infty \leq 1$. Let $\theta_\tau(w) = \mu_\tau^\top w$. By the induction condition $(i)$, we have that $X_\tau \succeq \frac{N}{8m} \check{\Lambda}_\tau$ for $\tau \in [h-1]$.

By Lemma 14 and the induction condition (iii) that $\lambda_{\tau,i}^2 \phi_{\tau,i}^\top \check{\Lambda}_\tau^{-1} \phi_{\tau,i} \le f_1$, with probability $1 - \frac{\delta}{16mH^2}$, we have that

$$\left| \phi^\top \theta_\tau(w) - \phi^\top X_\tau^{-1} \sum_{i=1}^N \lambda_{\tau,i}^2 \phi_{\tau,i} \cdot \left( \phi_{\tau,i}^\top \theta_\tau(w) + \epsilon_i \right) \right|$$

$$\le 8\sqrt{\phi^\top X_\tau^{-1} \phi \cdot d \log\left(\frac{dH}{\epsilon\delta}\right)} + 4\sqrt{\max_i \lambda_{\tau,i}^2 \phi_{\tau,i}^\top X_\tau^{-1} \phi_{\tau,i} \cdot \phi^\top X_\tau^{-1} \phi \cdot d \log\left(\frac{dH}{\epsilon\delta}\right)} + \zeta$$

$$\le 32\sqrt{\frac{md \log\left(\frac{dH}{\epsilon\delta}\right)}{N}} + \frac{32md\sqrt{f_1} \log\left(\frac{dH}{\epsilon\delta}\right)}{N} \tag{25}$$

for all $\phi$ such that $\|\phi\|_2 \le 1$ and $\phi^\top \check{\Lambda}_\tau^{-1} \phi \le 1$.

Let $\{v_\tau(s)\}$ and $\{v_\tau^*(s)\}$ denote respectively the value function under the policy $\pi^i$ and the optimal value function. Let $v_0 = \mathbb{E}_{s_1 \sim d_{\text{ini}}}[v_1(s_1)]$ and $v_0^* = \max_\pi \mathbb{E}_\pi[r_h^i(s_h)]$. Because $r_\tau^i(s,a) \in [0,1]$ for any proper $(s,a,\tau)$, we learn that $v_\tau(s), v_\tau^*(s), v_0, v_0^* \in [0,1]$. Recall the definition of $\{V_\tau(s)\}$ in Algorithm 5. We next prove by induction that $V_\tau(s) \ge v_\tau^*(s) \ge v_\tau(s)$ for any $s \in \mathcal{S}$ and $1 \le \tau \le h$. For $\tau = h$, the inequality is trivial. Assume $V_\tau(s) \ge v_\tau(s)$ for any $\ell \le \tau \le h$. By equation 25 with $w = V_\ell(\cdot)$

$$Q_{\ell-1}(s,a) \ge \mathbb{E}_{s' \sim P_{\ell-1,s,a}}[V_\ell(s')] \ge \mathbb{E}_{s' \sim P_{\ell-1,s,a}}[v_\ell^*(s')] \tag{26}$$

when $\phi_{\ell-1}^\top(s,a) \check{\Lambda}_{\ell-1}^{-1} \phi_{\ell-1}(s,a) \le 1$. In the case $\phi_{\ell-1}^\top(s,a) \check{\Lambda}_{\ell-1}^{-1} \phi_{\ell-1}(s,a) > 1$, we have that

$$Q_{\ell-1}(s,a) = \mathbb{E}_{s' \sim P_{\ell-1,s,a}}[V_\ell(s')] = 0 \tag{27}$$

because $P_{\ell-1,s,a} = \mathbf{1}_{\mathbf{z}}$.

Therefore, we have that

$$V_{\ell-1}(s) = \text{Range}_{[0,1]} \left( \max_a Q_{\ell-1}(s,a) \right) \ge \text{Range}_{[0,1]} \left( \max_a \mathbb{E}_{s' \sim P_{\ell-1,s,a}}[v_\ell^*(s')] \right) = v_{\ell-1}^*(s).$$

By Bernstein's inequality, with probability $1 - \frac{\delta}{16mH}$, it holds that

$$V_0 = \frac{1}{N} \sum_{i=1}^N V_1(s_{1,i}) + 2\sqrt{\frac{H^2 \log(1/\delta)}{N}} + 2\frac{H \log(16m/\delta)}{N} \ge \mathbb{E}_{s_1 \sim d_{\text{ini}}}[V_1(s_1)] \ge \mathbb{E}_{s_1 \sim d_{\text{ini}}}[v_1^*(s_1)] = v_0^*.$$

To bound the gap $\max_\pi \mathbb{E}_\pi[r_h^i(s_h)] - \mathbb{E}_{\pi^i}[r_h^i(s_h)]$, direct computation gives that

$$\max_\pi \mathbb{E}_\pi[r_h^i(s_h)] - \mathbb{E}_{\pi^i}[r_h^i(s_h)]$$

$$= v_0^* - \mathbb{E}_{\pi^i}[r_h^i(s_h)]$$

$$\le V_0^i - \mathbb{E}_{\pi^i}[r_h^{i-1}(s_h)]$$

$$= V_0^i - \mathbb{E}_{s_1 \sim d_{\text{ini}}}[V_1(s_1)] + \mathbb{E}_{\tau=1}^h \left[ V_\tau(s_\tau) - P_{\tau,s_\tau,a_\tau}^\top V_{\tau+1}(\cdot) \right]$$

$$\le 2\sqrt{\frac{H^2 \log(1/\delta)}{N}} + 2\frac{H \log(1/\delta)}{N} + 2\sum_{\tau=1}^h \left( 32\sqrt{\frac{md \log\left(\frac{dH}{\epsilon\delta}\right)}{N}} + \frac{32md\sqrt{f_1} \log\left(\frac{dH}{\epsilon\delta}\right)}{N} \right)$$

$$\tag{28}$$

$$= 2\sqrt{\frac{H^2 \log(1/\delta)}{N}} + 2\frac{H \log(1/\delta)}{N} + 2H \left( 32\sqrt{\frac{md \log\left(\frac{dH}{\epsilon\delta}\right)}{N}} + \frac{32md\sqrt{f_1} \log\left(\frac{dH}{\epsilon\delta}\right)}{N} \right)$$

$$= B,$$

where equation 28 is by plugging $\phi_{\tau,s_\tau,a_\tau} = \phi$ and $w = V_{\tau+1}(\cdot)$ into equation 25:

$$V_\tau(s_\tau) - P_{\tau,s_\tau,a_\tau}^\top V_{\tau+1}(\cdot) \le 2 \left( 32\sqrt{\frac{md \log\left(\frac{dH}{\epsilon\delta}\right)}{N}} + \frac{32md\sqrt{f_1} \log\left(\frac{dH}{\epsilon\delta}\right)}{N} \right).$$

$$\square$$

## C.5 STATEMENT AND PROOF OF LEMMA 14

**Lemma 14.** *Fix , $v \in \mathbb{R}^{\mathcal{S}}$ such that $\|v\|_\infty \leq 1$ and $f : \mathcal{S} \to \mathbb{R}^{d^2}$ such that $0 \preceq f(s) \preceq \Lambda, \forall s \in \mathcal{S}$ for some $\Lambda$. Let $\{\phi_{\tau,i}, \tilde{s}_{\tau,i}, \lambda_{\tau,i}\}_{i=1}^N$ be a dataset independent of $v$ and $f$ from the $\tau$-th layer. Let $X_\tau = \sum_{i=1}^N \lambda_{\tau,i}^2 \phi_{\tau,i} \phi_{\tau,i}^\top + z\mathbf{I}$. With probability $1 - \frac{\delta}{16mH^2}$, it holds that*

$$
\left| \phi^\top \theta(v) - \phi^\top X_\tau^{-1} \sum_{i=1}^N \phi_{\tau,i} v(\tilde{s}_{\tau,i}) \right|
$$

$$
\leq 8\sqrt{\phi^\top X_\tau^{-1} \phi (d \log(\frac{dH}{\epsilon\delta})} + 4\sqrt{\max_i \phi_{\tau,i}^\top X_\tau^{-1} \phi_{\tau,i} \phi^\top X_\tau^{-1} \phi} \cdot d\log(\frac{dH}{\epsilon\delta}) + \zeta.
$$

*and*

$$
\tag{29}
$$

$$
\left| \phi^\top \mu^\top f - \phi^\top X_\tau^{-1} \sum_{i=1}^N \lambda_{\tau,i}^2 \phi_{\tau,i} f(\tilde{s}_{\tau,i}) \right|
$$

$$
\preceq \left( 16\sqrt{\phi^\top X_\tau^{-1} \phi d \log(\frac{dH}{\epsilon\delta})} + 8\sqrt{\max_i \phi_{\tau,i}^\top X_\tau^{-1} \phi_{\tau,i} \phi^\top X_\tau^{-1} \phi} d\log(\frac{dH}{\epsilon\delta}) + \zeta \right) \Lambda.
$$

*for any $\phi$ such that $\|\phi\|_2 \leq 1$.*

*Proof.* Let $\Phi(\xi)$ be an $\xi$-net of the $d$-dimensional unit ball w.r.t. $L_2$ norm. Recall that $\xi = \left( \frac{\epsilon}{10d^2H^2} \right)^{10}$. Then $\log(\xi) \leq 20\log(dH/\epsilon)$. Let

$$
\mathcal{E}_1(\phi, v)
$$

$$
:= \left\{ \left| \phi^\top \theta(v) - \phi^\top X_\tau^{-1} \sum_{i=1}^N \lambda_{\tau,i}^2 \phi_{\tau,i} v(\tilde{s}_{\tau,i}) \right| \leq 4\sqrt{\phi^\top X_\tau^{-1}\phi \log(1/\delta)} + 2\sqrt{\max_i \phi_{\tau,i}^\top X_\tau^{-1} \phi_{\tau,i} \phi^\top X_\tau^{-1}\phi} \cdot \log(1/\delta) \right\}.
$$

Then $\Pr[\mathcal{E}(\phi, v)] \leq 2\delta$ by Bernstein's inequality. Assume $\cup_{\phi \in \Phi(\xi)} \mathcal{E}_1(\phi, v)$ holds. Then for any $\phi \in \mathbb{R}^d$, letting $\psi$ be the nearest neighbor of $\phi$ in $\Phi(\xi)$, it holds that

$$
\left| \phi^\top \theta(v) - \phi^\top X_\tau^{-1} \sum_{i=1}^N \phi_{\tau,i} v(\tilde{s}_{\tau,i}) \right|
$$

$$
\leq \left| \phi^\top \theta(v) - \psi^\top \theta(v) \right| + \left| \phi^\top X_\tau^{-1} \sum_{i=1}^N \phi_{\tau,i} v(\tilde{s}_{\tau,i}) - \psi^\top X_\tau^{-1} \sum_{i=1}^N \phi_{\tau,i} v(\tilde{s}_{\tau,i}) \right| + \left| \psi^\top \theta(v) - \psi^\top X_\tau^{-1} \sum_{i=1}^N \phi_{\tau,i} v(\tilde{s}_{\tau,i}) \right|
$$

$$
\leq \xi + \frac{N\xi}{z} + 4\sqrt{\psi^\top X_\tau^{-1}\psi \log(1/\delta)} + 2\sqrt{\max_i \phi_{\tau,i}^\top X_\tau^{-1}\phi_{\tau,i}\psi^\top X_\tau^{-1}\psi} \cdot \log(1/\delta)
$$

$$
\leq 4\sqrt{\phi^\top X_\tau^{-1}\phi \log(1/\delta)} + 2\sqrt{\max_i \phi_{\tau,i}^\top X_\tau^{-1}\phi_{\tau,i}\phi^\top X_\tau^{-1}\phi} \cdot \log(1/\delta) + \xi + \frac{N\xi}{z} + 6\log(1/\delta)\frac{2\xi}{z\sqrt{z}}
$$

$$
\leq 4\sqrt{\phi^\top X_\tau^{-1}\phi \log(1/\delta)} + 2\sqrt{\max_i \phi_{\tau,i}^\top X_\tau^{-1}\phi_{\tau,i}\phi^\top X_\tau^{-1}\phi} \cdot \log(1/\delta) + \zeta.
$$

Noting that $|\Phi(\xi)| \leq (d/\xi)^d$, we have that $\Pr[\cup_{\phi \in \Phi(\xi)}]\mathcal{E}_1(\phi, v) \leq 2(d/\xi)^d\delta$. By replacing $\delta$ with $\frac{\delta}{16mH|\Phi(\xi)|}$, with probability $1 - 2\delta$, it holds that

$$
\left| \phi^\top \theta(v) - \phi^\top X_\tau^{-1} \sum_{i=1}^N \phi_{\tau,i} v(\tilde{s}_{\tau,i}) \right|
$$

$$
\leq 4\sqrt{\phi^\top X_\tau^{-1}\phi (d + \log(\frac{d}{\xi\delta})} + 2\sqrt{\max_i \phi_{\tau,i}^\top X_\tau^{-1}\phi_{\tau,i}\phi^\top X_\tau^{-1}\phi} \cdot (d + \log(\frac{d}{\xi\delta})) + \zeta.
$$

$$
\tag{30}
$$

for any $\phi$ such that $\|\phi\|_2 \le 1$.

Define $\mathcal{E}_2(\phi, f)$ to be the event where

$$\left| \phi^\top \mu_\tau^\top f - \phi^\top X_\tau^{-1} \sum_{i=1}^N \lambda_{\tau,i}^2 \phi_{\tau,i} f(\tilde{s}_{\tau,i}) \right| \preceq \left( 4\sqrt{\phi^\top X_\tau^{-1} \phi \log(\frac{1}{\delta})} + 2\sqrt{\max_i \phi_{\tau,i}^\top X_\tau^{-1} \phi_{\tau,i} \phi^\top X_\tau^{-1} \phi} \log(\frac{1}{\delta}) \right) \Lambda \tag{31}$$

holds. We then show that $\Pr[\mathcal{E}_2(\phi, f)] \le 2\delta$.

$$\phi^\top \mu_\tau^\top f - \phi^\top X_\tau^{-1} \sum_{i=1}^N \lambda_{\tau,i}^2 \phi_{\tau,i} f(\tilde{s}_{\tau,i}) = \phi^\top X_\tau^{-1} X_\tau \mu_\tau^\top f - \phi^\top X_\tau^{-1} \sum_{i=1}^N \lambda_{\tau,i}^2 \phi_{\tau,i} f(\tilde{s}_{\tau,i})$$

$$= \phi^\top X_\tau^{-1} \left( X_\tau \mu_\tau^\top f - \sum_{i=1}^N \lambda_{\tau,i}^2 \phi_{\tau,i} \left( \phi_{\tau,i} \mu_\tau^\top f + \epsilon_{\tau,i} \right) \right)$$

$$= -\sum_{i=1}^N \phi^\top X_\tau^{-1} \lambda_{\tau,i}^2 \phi_{\tau,i} \epsilon_{\tau,i} + \phi^\top X_\tau^{-1} z \mu_\tau^\top f, \tag{32}$$

where we define $\epsilon_{\tau,i} = \mathbb{E}_{s' \sim P_{\tau,s,a}}[f(s')] - f(\tilde{s}_{\tau,i})$ with $(s, a)$ being the state-action pair such that $\phi_\tau(s, a) = \phi_{\tau,i}$. Noting that $-\Lambda \preceq \epsilon_{\tau,i} \preceq \Lambda$ with probability 1, we have that

$$\sum_{i=1}^N \phi^\top X_\tau^{-1} \lambda_{\tau,i}^2 \phi_{\tau,i} \epsilon_{\tau,i} \preceq 2\sqrt{\log(d/\delta) \cdot \sum_{i=1}^N \left( \lambda_{\tau,i}^2 \phi^\top X_\tau^{-1} \phi_{\tau,i} \right)^2 \Lambda} + 2\max_i \left| \lambda_{\tau,i}^2 \phi^\top X_\tau^{-1} \phi_{\tau,i} \right| \log(d/\delta) \Lambda$$

$$\preceq 2\sqrt{\log(d/\delta) \phi^\top X_\tau^{-1} \phi} \Lambda + 2\max_i \sqrt{\phi^\top X_\tau^{-1} \phi \cdot \lambda_{\tau,i}^2 \phi_{\tau,i}^\top X_\tau^{-1} \phi_{\tau,i}} \Lambda \tag{33}$$

holds with probability $1 - \delta$. In a similar way, with probability $1 - \delta$, we have

$$\tag{34}$$

$$-\sum_{i=1}^N \phi^\top X_\tau^{-1} \lambda_{\tau,i}^2 \phi_{\tau,i} \epsilon_{\tau,i} \preceq 2\sqrt{\log(d/\delta) \phi^\top X_\tau^{-1} \phi} \Lambda + 2\max_i \sqrt{\phi^\top X_\tau^{-1} \phi \cdot \lambda_{\tau,i}^2 \phi_{\tau,i}^\top X_\tau^{-1} \phi_{\tau,i}} \Lambda. \tag{35}$$

To bound the second term $z\phi^\top X_\tau^{-1} \mu_\tau^\top f$ in equation 32, we have

$$|z\phi^\top X_\tau^{-1} \mu_\tau^\top v| \le z\|\phi^\top X_\tau^{-1}\|_2 \|\mu_\tau^\top v\|_2$$

$$\le \sqrt{z}\sqrt{z\phi^\top X_\tau^{-2} \phi} \cdot \sqrt{d}$$

$$\le \sqrt{zd \cdot \phi^\top X_\tau^{-1} \phi}$$

$$\le \sqrt{\phi^\top X_\tau^{-1} \phi} \tag{36}$$

for any $v \in \mathbb{R}^S$ such that $\|v\|_\infty \le 1$. As a result, we have $\|z\phi^\top X_\tau^{-1} \mu_\tau^\top\|_1 \le \sqrt{\phi^\top X_\tau^{-1} \phi}$. Noting that $0 \preceq f(s) \preceq \Lambda$ for all $s \in \mathcal{S}$, we have that

$$-\sqrt{\phi^\top X_\tau^{-1} \phi} \Lambda \preceq z\phi^\top X_\tau^{-1} \mu_\tau^\top f \preceq \sqrt{\phi^\top X_\tau^{-1} \phi} \Lambda. \tag{37}$$

By equation 32, equation 33, equation 35 and equation 37, we have that

$$\left| \phi^\top \mu_\tau^\top f - \phi^\top X_\tau^{-1} \sum_{i=1}^N \lambda_{\tau,i}^2 \phi_{\tau,i} f(\tilde{s}_{\tau,i}) \right|$$

$$\preceq 4\sqrt{\log(d/\delta) \phi^\top X_\tau^{-1} \phi} \Lambda + 2\max_i \sqrt{\phi^\top X_\tau^{-1} \phi \cdot \lambda_{\tau,i}^2 \phi_{\tau,i}^\top X_\tau^{-1} \phi_{\tau,i}} \Lambda \tag{38}$$

The proof is finished. Assume $\cup_{\phi \in \Phi(\xi)} \mathcal{E}_2(\phi, f)$ holds. Fix $\phi$ and let $\psi$ be the nearest neighbor of $\phi$ in $\Phi(\xi)$. We then have that

$$\phi^\top \mu_\tau^\top f - \phi^\top X_\tau^{-1} \sum_{i=1}^N \phi_{\tau,i} f(\tilde{s}_{\tau,i})$$

$$= \left( \phi^\top \mu_\tau^\top f - \psi^\top \mu_\tau^\top f \right) + \left( \phi^\top X_\tau^{-1} \sum_{i=1}^N \phi_{\tau,i} f(\tilde{s}_{\tau,i}) - \psi^\top X_\tau^{-1} \sum_{i=1}^N \phi_{\tau,i} f(\tilde{s}_{\tau,i}) \right)$$

$$+ \left( \psi^\top \theta(v) - \psi^\top X_\tau^{-1} \sum_{i=1}^N \phi_{\tau,i} f(\tilde{s}_{\tau,i}) \right). \quad (39)$$

We then bound the three terms in equation 39 separately. For the first term, we have that $|(\phi - \psi)^\top \mu_\tau^\top v| \leq \xi\sqrt{d}$ for any $v \in \mathbb{R}^{\mathcal{S}}$ such that $\|v\|_\infty \leq 1$. As a result, we have that $\|\mu_\tau(\phi - \psi)\|_1 \leq \xi\sqrt{d}$, which implies that

$$-\xi\sqrt{d}\Lambda \preceq \phi^\top \mu_\tau^\top f - \psi^\top \mu_\tau^\top f \preceq \xi\sqrt{d}\Lambda. \quad (40)$$

For the second term, we have that

$$\left| \phi^\top X_\tau^{-1} \sum_{i=1}^N \phi_{\tau,i} v(\tilde{s}_{\tau,i}) - \psi^\top X_\tau^{-1} \sum_{i=1}^N \phi_{\tau,i} v(\tilde{s}_{\tau,i}) \right| \leq \frac{N\xi}{z}$$

for any $v \in \mathbb{R}^{\mathcal{S}}$ such that $\|v\|_\infty \leq 1$. Using similar arguments, we learn that $\left\| \phi^\top X_\tau^{-1} \sum_{i=1}^N \phi_{\tau,i} - \psi^\top X_\tau^{-1} \sum_{i=1}^N \phi_{\tau,i} \right\|_1 \leq \frac{\sqrt{d}N\xi}{z}$ and

$$-\frac{\sqrt{d}N\xi}{z}\Lambda \preceq \phi^\top X_\tau^{-1} \sum_{i=1}^N \phi_{\tau,i} f(\tilde{s}_{\tau,i}) - \psi^\top X_\tau^{-1} \sum_{i=1}^N \phi_{\tau,i} f(\tilde{s}_{\tau,i}) \preceq \frac{\sqrt{d}N\xi}{z}\Lambda. \quad (41)$$

By $\cup_{\phi \in \Phi(\xi)} \mathcal{E}_2(\phi, f)$, we could bound the third term as

$$\left| \psi^\top \theta(v) - \psi^\top X_\tau^{-1} \sum_{i=1}^N \phi_{\tau,i} f(\tilde{s}_{\tau,i}) \right| \preceq 4\sqrt{\log(d/\delta)\psi^\top X_\tau^{-1}\psi}\Lambda + 2\max_i \sqrt{\psi^\top X_\tau^{-1}\psi\lambda_{\tau,i}^2 \phi_{\tau,i}^\top X_\tau^{-1}\phi_{\tau,i}}\Lambda. \quad (42)$$

Putting equation 40, equation 41 and equation 42 together, we learn that

$$\left| \phi^\top \mu_\tau^\top f - \phi^\top X_\tau^{-1} \sum_{i=1}^N \phi_{\tau,i} f(\tilde{s}_{\tau,i}) \right|$$

$$\preceq \left( \xi\sqrt{d} + \frac{\sqrt{d}N\xi}{z} + 4\sqrt{\log(d/\delta)\psi^\top X_\tau^{-1}\psi} + 2\max_i \sqrt{\psi^\top X_\tau^{-1}\psi\lambda_{\tau,i}^2 \phi_{\tau,i}^\top X_\tau^{-1}\phi_{\tau,i}} \right) \Lambda$$

$$\leq \left( \xi\sqrt{d} + \frac{\sqrt{d}N\xi}{z} + \frac{12\log(d/\delta)\xi}{z\sqrt{z}} + 4\sqrt{\log(d/\delta)\phi^\top X_\tau^{-1}\phi} + 2\max_i \sqrt{\phi^\top X_\tau^{-1}\phi\lambda_{\tau,i}^2 \phi_{\tau,i}^\top X_\tau^{-1}\phi_{\tau,i}} \right) \Lambda$$

$$\leq \left( 4\sqrt{\log(d/\delta)\phi^\top X_\tau^{-1}\phi} + 2\max_i \sqrt{\phi^\top X_\tau^{-1}\phi\lambda_{\tau,i}^2 \phi_{\tau,i}^\top X_\tau^{-1}\phi_{\tau,i}} + \zeta \right) \Lambda. \quad (43)$$

The proof is finished by replacing $\delta$ with $\frac{\delta}{16mH|\Phi(\xi)|}$.

$\square$

## C.6 PROOF OF LEMMA 5

Let $\Theta$ be an $\frac{\epsilon}{8dH}$-net of $\mathbb{B}_2(\sqrt{d})^H$. Without loss of generality, we can take $\Theta$ to be the $dH$-dimensional grid with distance $\frac{\epsilon}{8dH}$. Let $\mathrm{Proj}_\Theta(\cdot)$ be the projection function to $\Theta$ by projecting

each dimension to the grid. It is obvious that if $\theta = \{\theta_h\}_{h \in [H]}$ satisfies that $\|\theta_h\|_2 \leq d$ for each $h$, $\|\text{Proj}_{\Theta,h}(\theta)\|_2 \leq 2d$.

It suffices to show that for any kernel $\{\theta_h\}_{h \in [H]} \in \Theta$, the output policy is $\frac{3}{4}\epsilon$-optimal. Assume the conditions in Lemma 6 holds. Let $\check{M}$ be the final truncated MDP $M_H$. Then we have that

$$\max_\pi \Pr_\pi \left[ \exists h \in [H], \phi_h^\top \check{\Lambda}_h \phi_h > 1 \right] \leq H \cdot \frac{\epsilon}{8H^2} \leq \frac{\epsilon}{8H}.$$

As a result, for any $\pi$ and reward function $r$ such that $\|r\|_\infty \leq 1$, we have that $|\mathbb{E}_\pi[\sum_{h=1}^H r_h] - \mathbb{E}_{\pi,\check{M}}[\sum_{h=1}^H r_h]| \leq \frac{\epsilon}{8}$.

Fix reward kernel $\theta = \{\theta_h\}_{h \in [H]} \in \Theta$. We continue the analysis by assuming the ground MDP is $\check{M}$. Let $\pi$ be the returned policy and $\pi^*$ be the optimal policy. Let $\{V_{h,\theta}^*(s), Q_{h,\theta}^*(s,a)\}$ and $\{V_{h,\theta}^\pi(s), Q_{h,\theta}^\pi(s,a)\}$ be respectively the optimal value function and the value function of $\pi$. In particular, we use $V_{0,\theta}^*$ ($V_\theta^\pi$) to denote the value of the optimal policy ($\pi$). Let $\{V_{h,\theta}(s), Q_{h,\theta}(s,a)\}$ be the value of $\{V_h(s), Q_h(s,a)\}$ in Algorithm 4 with input kernel as $\theta$. Let $V_{0,\theta} = \mathbb{E}_{s_1 \sim d_{\text{ini}}}[V_{1,\theta}(s_1)]$. When $\theta$ is clear from the context, we omit $\theta$ in the subscript.

We then have that

$$V_0^* - V_0^\pi = (V_0^* - V_0) + (V_0 - V_0^\pi). \tag{44}$$

We then prove by induction that $V_h^*(s) - V_h(s) \leq (H-h) \cdot \frac{\epsilon}{8H}$ for all $s \in \mathcal{S}$ and $h \in [H]$. The inequality is trivial for $h = H$. Now we assume it is correct for all $h \geq \ell$. Let $X_\tau = \sum_{i=1}^N \lambda_{\tau,i}^2 \phi_{\tau,i} \phi_{\tau,i}^\top + z\mathbf{I}$ for $\tau \in [H]$. Recall that $\Phi(\xi)$ is an $\xi$-net of the $d$-dimensional unit ball. Fix $\phi \in \Phi(\xi)$ with $\|\phi\|_2 \leq 1$ and $V \in \mathbb{R}^\mathcal{S}$ with $\|V\|_\infty \leq H$. By Bernstein's inequality, with probability $1 - \frac{\delta}{4H|\Phi(\xi)| \cdot |\Theta|}$, it holds that

$$\left| \phi^\top X_h^{-1} \sum_{i=1}^N \lambda_{h,i}^2 \phi_{h,i} V(\tilde{s}_{h,i}) - \phi^\top \mu_\tau^\top V \right|$$

$$\leq 4\sqrt{\phi^\top X_\tau^{-1} \phi \log\left(\frac{4H|\Phi(\xi)| \cdot |\Theta|}{\delta}\right)} + 2\max_i \sqrt{\phi^\top X_h^{-1} \phi \cdot \lambda_{h,i}^2 \phi_{h,i}^\top X_h^{-1} \phi_{h,i} \log\left(\frac{4H|\Phi(\xi)| \cdot |\Theta|}{\delta}\right)}$$

$$\leq \sqrt{\frac{128m}{N} \log\left(\frac{4H|\Phi(\xi)| \cdot |\Theta|}{\delta}\right)} + \sqrt{\frac{32m}{N} \cdot \phi^\top X_h^{-1} \phi \log\left(\frac{4H|\Phi(\xi)| \cdot |\Theta|}{\delta}\right)}.$$

With a union bound over $\phi \in \Phi(\xi)$, we learn that, with probability $1 - \frac{\delta}{4H|\Theta|}$,

$$\left| \phi^\top X_h^{-1} \sum_{i=1}^N \lambda_{h,i}^2 \phi_{h,i} V(\tilde{s}_{h,i}) - \phi^\top \mu_h^\top V \right| \leq 32\sqrt{\frac{mdH \log\left(\frac{dH}{\epsilon\delta}\right)}{N}} + \sqrt{\frac{128m}{N} \cdot \phi^\top X_h^{-1} \phi} \cdot dH \log\left(\frac{dH}{\epsilon\delta}\right)$$

$$\leq 32\sqrt{\frac{mdH \log\left(\frac{dH}{\epsilon\delta}\right)}{N}} + \frac{32mdH \log\left(\frac{dH}{\epsilon\delta}\right)}{N}$$

$$\leq \frac{\epsilon}{16H}$$

for any $\phi$ such that $\|\phi\|_2 \leq 1$ and $\phi^\top \check{\Lambda}_h \phi \leq 1$. Note that $V_{h+1,\theta}(\cdot)$ is determined by $\theta = \{\theta_h\}_{h \in [H]}$ and the sub-datasets after the $h$-th layer (non-inclusive). With a union bound over $\theta \in \Theta$, we learn that: with probability $1 - \frac{\delta}{4}$,

$$\left| \phi^\top X_h^{-1} \sum_{i=1}^N \lambda_{h,i}^2 \phi_{h,i} V_{h+1,\theta}(\tilde{s}_{h,i}) - \phi^\top \mu_h^\top V_{h+1,\theta} \right| \leq \frac{\epsilon}{16H} \tag{45}$$

for any $\phi$ such that $\|\phi\|_2 \leq 1$, $\phi^\top \check{\Lambda}_h \phi \leq 1$ and $\theta \in \Theta$. Then we have that

$$V_{\ell-1}^*(s) - V_{\ell-1}(s)$$
$$= Q_{\ell-1}^*(s, \pi_{\ell-1}^*(s)) - V_{\ell-1}(s)$$
$$\leq Q_{\ell-1}^*(s, \pi_{\ell-1}^*(s)) - Q_{\ell-1}(s, \pi_{\ell-1}^*(s))$$
$$\leq P_{\ell-1,s,\pi_{\ell-1}^*(s)}^\top(V_\ell^* - V_\ell) + P_{\ell-1,s,\pi_{\ell-1}^*(s)}^\top V_\ell - \phi_{\ell-1,s,\pi_{\ell-1}^*}^\top X_{\ell-1}^{-1} \sum_{i=1}^N \lambda_{\ell-1}^2 \phi_{\ell-1,i} V_\ell(\tilde{s}_{\ell,i}) + \frac{\epsilon}{16H}$$

$$\tag{46}$$

$$\leq P_{\ell-1,s,\pi_{\ell-1}^*(s)}^\top(V_\ell^* - V_\ell) + \frac{\epsilon}{8H}$$
$$\leq \frac{\epsilon(H-h)}{8H}.$$

As a result, we learn that $V_0^* - V_0 \leq \frac{\epsilon}{8}$. For the second term $(V_0 - V_0^\pi)$ in equation 44, using similar arguments, we have that

$$V_0 - V_0^\pi = \mathbb{E}_\pi\left[\sum_{h=1}^H Q_h(s_h, a_h) - \phi_h^\top \theta_h - P_{h,s_h,a_h}^\top V_{h+1}(s_h)\right]$$
$$\leq H \cdot \frac{\epsilon}{8H}$$
$$\leq \frac{\epsilon}{8}.$$

$$\tag{47}$$

Putting all together, with probability $1 - \frac{\delta}{2}$, we have that $V_{0,\theta}^* - V_{0,\theta}^\pi \leq \frac{\epsilon}{4} \leq \frac{5\epsilon}{8}$ for all $\theta \in \Theta$. As a result, $\pi$ is at least a $\frac{3}{4}\epsilon$-optimal policy under the original $MDP$ $M$. The proof is completed.

### C.7 STATEMENT AND PROOF OF LEMMA 15

**Lemma 15.** *By running Algorithm 3, we have the following claims: (1) The iteration in line 3 ends in $10d \log\left(\frac{2x}{v} + 1\right)$ rounds; (2) Let $\Lambda_{\text{end}}$ be the final value of $\Lambda$. Then it holds that*

$$\Pr_\pi\left[\phi_h^\top(\Lambda_{\text{end}})^{-1}\phi_h > f_1\right] \leq \frac{d}{f_1(1 - 3Hx)}.$$

*Proof.* Fix $\pi$. Let $\hat{F}_0(\Lambda)$ be the value of $\hat{F}_0$ computed with truncation matrix as $\Lambda$ in line 12-24 in Algorithm 3. Let $F_0(\Lambda) := \mathbb{E}_\pi\left[\mathrm{T}(\phi_h\phi_h^\top, f_1\Lambda)\right]$.

**Number of iterations.** Let $\Lambda_i$ be the value of $\Lambda$ after the $i$-th iteration. Suppose there are $m$ iterations. For $1 \leq i \leq m$, we have that $\Lambda_i = \hat{F}_0(\Lambda_{i-1})$ satisfies that

$$(1 - 3Hx)F_0(\Lambda_{i-1}) \preceq \Lambda_i \preceq (1 + 3Hx)F_0(\Lambda_{i-1}) + 3Hx\Lambda_{i-1} + 3H\zeta\mathbf{I} \preceq (1 + 6Hx)\Lambda_{i-1} + 3H\zeta\mathbf{I}.$$

$$\tag{48}$$

By the update rule, we learn that

$$\Lambda_i \preceq (1 + 6Hx)\Lambda_{i-1} + 3H\zeta\mathbf{I};$$
$$\Lambda_i + \frac{\zeta}{2x}\zeta\mathbf{I} \not\succeq \frac{1}{2}\Lambda_{i-1},$$

Let $\check{\Lambda}_i = \Lambda_i + \frac{\zeta}{2x}\mathbf{I}$ for $i \geq 0$. Then we learn that

$$\check{\Lambda}_i \preceq (1 + 6Hx)\check{\Lambda}_{i-1}, \qquad \check{\Lambda}_i \not\succeq \frac{1}{2}\check{\Lambda}_{i-1}, \qquad \check{\Lambda}_i \succeq \frac{\zeta}{2x}\mathbf{I}.$$

As a result, the maximal eigenvalue of $\check{\Lambda}_{i-1}^{-1/2}\check{\Lambda}_i\check{\Lambda}_{i-1}^{-1/2}$ is at most $(1+6Hx)$, while the minimal eigenvalue of $\check{\Lambda}_{i-1}^{-1/2}\check{\Lambda}_i\check{\Lambda}_{i-1}^{-1/2}$ is at most $\frac{1}{2}$. Then we have that

$$\log(\det(\check{\Lambda}_i)) - \log(\det(\check{\Lambda}_{i-1})) + d\log(1+6Hx) - \log(2) \leq -\frac{1}{10}. \qquad (49)$$

By noting that $d\log(\zeta/2x) \leq \log(\det(\check{\Lambda}_i))$ and $\log(\det(\check{\Lambda}_0)) \leq d\log(1+\zeta/2x)$, we learn that $m \leq 10d\log\left(\frac{2x}{\zeta}+1\right) \leq f_8$. Let $\Lambda_{\mathrm{end}} = \Lambda_m$.

**Truncation probability.** Note that $\Lambda_{\mathrm{end}} \succeq (1-3Hx)F_0(\Lambda_{\mathrm{end}})$ and $F_0(\Lambda_{\mathrm{end}}) = \mathbb{E}_\pi\left[\mathtt{T}(\phi_h\phi_h^\top, f_1\Lambda_{\mathrm{end}})\right]$. We then have that

$$\mathbb{E}_\pi\left[\mathrm{Trace}\left(\mathtt{T}(\phi_h\phi_h^\top, f_1\Lambda_{\mathrm{end}})(\Lambda_{\mathrm{end}})^{-1}\right)\right] \leq \frac{d}{(1-3Hx)}.$$

On the other hand, by noting that

$$\mathrm{Pr}_\pi\left[\phi_h^\top(\Lambda_{\mathrm{end}})^{-1}\phi_h > f_1\right] \cdot f_1 \leq \mathbb{E}_\pi\left[\mathrm{Trace}\left(\mathtt{T}(\phi_h\phi_h^\top, f_1\Lambda_{\mathrm{end}})(\Lambda_{\mathrm{end}})^{-1}\right)\right] \leq \frac{d}{(1-3Hx)},$$

we have

$$\mathrm{Pr}_\pi\left[\phi_h^\top(\Lambda_{\mathrm{end}})^{-1}\phi_h > f_1\right] \leq \frac{d}{f_1(1-3Hx)}.$$

$\square$

## C.8 STATEMENT AND PROOF OF LEMMA 16

**Lemma 16.** *Recall that $z = \frac{100000\epsilon^2}{d^2H^5}$. Let $\mathcal{D}_h = \{\phi_{h,i}, \tilde{s}_{h,j}, \lambda_{h,i}\}_{i=1}^N$ be the one dataset in in Line 9, Algorithm 6. With probability $1 - \frac{\delta}{16m^2H^2}$, it holds that*

$$\sum_{i=1}^N \lambda_{h,i}^2\phi_{h,i}\phi_{h,i}^\top + z\mathbf{I} \succeq \frac{N}{8m}\cdot\check{\Lambda}_h.$$

*Proof.* Let $X_h^i$ and $Y_h^i$ be respectively the final value of $\Lambda$ and $\hat{F}_0$ in the $i$-th call of Algorithm 3 in the $h$-th round. It then holds that

$$(1+3Hx)\mathbb{E}_{\pi^{i,h}}\left[\mathtt{T}(\phi_h\phi_h^\top, f_1X_h^i)\right] + 3HxX_h^i + 3H\zeta\mathbf{I} + \frac{\zeta}{2x}\mathbf{I} \succeq Y_h^i + \frac{\zeta}{2x}\mathbf{I} \succeq \frac{1}{2}X_h^i$$

and

$$(1+3Hx)\mathbb{E}_{\pi^{i,h}}\left[\mathtt{T}(\phi_h\phi_h^\top, f_1X_h^i)\right] + 3Hx\left(2Y_h^i + \frac{\zeta}{x}\mathbf{I}\right) + 3H\zeta\mathbf{I} + \frac{\zeta}{2x}\mathbf{I} \succeq Y_h^i + \frac{\zeta}{2x}\mathbf{I}.$$

Because $\check{\Lambda}_h \succeq \frac{1}{2}X_h^i$

$$\mathbb{E}\left[\sum_{i=1}^N \lambda_{h,i}^2\phi_{h,i}\phi_{h,i}^\top\right] \succeq \frac{N}{2m}\sum_{j=1}^m \mathbb{E}_{\pi^{j,h}}\left[\mathtt{T}(\phi_h\phi_h^\top, f_1X_h^j)\right]$$

$$\succeq \frac{N}{2m}\cdot\sum_{j=1}^m\frac{1}{1+3Hx}\cdot\left((1-6Hx)Y_h^j + \frac{\zeta}{2x}\mathbf{I} - 6H\zeta\mathbf{I} - \frac{\zeta}{2x}\mathbf{I}\right)$$

$$\succeq \frac{N}{2m}\cdot\sum_{j=1}^m\left(\frac{1}{2}\bar{\Lambda}_h^j - \left(6H+\frac{1}{2x}\right)\zeta\mathbf{I}\right)$$

$$= \frac{N}{2m}\cdot\left(\frac{1}{2}\check{\Lambda}_h - \left(6H+\frac{1}{2x}\right)\zeta\mathbf{I} - \zeta\mathbf{I}\right). \qquad (50)$$

Also noting that $\lambda_{h,i}\phi_{h,i}\phi_{h,i}^\top \preceq f_1 \check{\Lambda}_h$ with probability 1, using Lemma 9, we have that, with probability $1 - \frac{\delta}{16mH^2}$,

$$\sum_{i=1}^N \lambda_{h,i}^2 \phi_{h,i}\phi_{h,i}^\top \succeq \frac{1}{2}\mathbb{E}\left[\sum_{i=1}^N \lambda_{h,i}^2 \phi_{h,i}\phi_{h,i}^\top\right] - f_1 \check{\Lambda}_h \log(16mH^2/\delta)$$

$$\succeq \frac{N}{8m}\check{\Lambda}_h - \frac{N\left(7H + \frac{1}{2x}\right)}{4m}\zeta\mathbf{I}$$

$$\succeq \frac{N}{8m}\check{\Lambda}_h - z\mathbf{I} \tag{51}$$

$\square$

## D  MISSING ALGORITHMS

In this section, we present the missing algorithms.

**Planning (Algorithm 4).**   This algorithm is used to compute the optimal policy given a group of datasets. The planning method is based on classical linear regression.

**Opt (Algorithm 5).**   This algorithm is used to compute the near-optimal policy given a fixed reward function. The planning method is based on classical linear regression.

**Policy-Execution (Algorithm 5).**   This algorithm is used to collect multiple copies of the datasets. The efficiency of the collected dataset is explained in Lemma 16.

---

**Algorithm 4** Planning

**Input**: reward kernel $\{\theta_h\}_{h\in[H]}$, dataset $\{\phi_{h,i}, \tilde{s}_{h,i}, \lambda_{h,i}\}_{i=1}^N\}_{h\in[H]}$ and block matrix $\{\check{\Lambda}_h\}_{h\in[H]}$;

$\{\theta_h\}_{h\in[H]} \leftarrow \mathrm{Proj}_\times(\{\theta_h\}_{h\in[H]})$;
$V_{H+1}(s) \leftarrow 0$ for all $s \in \mathcal{S}$;
**for** $h = H, H-1, \ldots, 1$ **do**
  **for** $(s,a) \in \mathcal{S} \times \mathcal{A}$; **do**
    $\phi \leftarrow \phi_h(s,a)$
    $Q_h(s,a) \leftarrow \begin{cases} \phi^\top\theta_h + \phi^\top\left(\sum_{i=1}^N \lambda_{h,i}^2\phi_{h,i}\phi_{h,i}^\top + z\mathbf{I}\right)^{-1}\sum_{i=1}^N \lambda_{h,i}^2\phi_{h,i}V_{H+1}(\tilde{s}_{h,i}), & \phi^\top\check{\Lambda}_h^{-1}\phi \leq 1; \\ 0, & \text{else}; \end{cases}$
    $Q_h(s,a) \leftarrow \mathrm{Range}_{[0,H]}(Q_h(s,a))$;
  **end for**
  **for** $s \in \mathcal{S}$ **do**
    $V_h(s) \leftarrow \max_a Q_h(s,a)$;
    $\pi_h(s) \leftarrow \arg\max_a Q_h(s,a)$;
  **end for**
**end for**
**return:** $\pi \leftarrow \{\pi_h\}_{h\in[H]}$.

---

## E  MISSING ALGORITHM AND PROOFS FOR THE FIRST LAYER

In this section, we propose the algorithm Ini-Sampling to collect the samples for the first layer. Below we prove that, by running Ini-Sampling, the three conditions in Lemma 6 holds for the first layer.

---

**Algorithm 5** Opt

---

**Input:** horizon $h$, reward function $r$, dataset $\{\phi_{\tau,i}, \tilde{s}_{\tau,i}, \lambda_{\tau,i}\}_{1 \leq i, 1 \leq \tau \leq h-1} \cup \{s_{1,i}\}_{i=1}^{N}$;

$V_h(s) \leftarrow \max_a r_h(s,a), \forall s \in \{\tilde{s}_{h-1,i}\}_{i \geq 1}$;

**for** $\tau = h-1, h-2, \ldots, 1$ **do**

    $X_\tau \leftarrow \sum_{i=1}^{N} \lambda_{\tau,i}^2 \phi_{\tau,i} \phi_{\tau,i}^\top + z\mathbf{I}$;

    **for** $s \in \{\tilde{s}_{\tau-1,i}\}_{i \geq 1}, a \in \mathcal{A}$ **do**

        $\phi \leftarrow \phi_\tau(s,a)$;

$$Q_\tau(s,a) \leftarrow \begin{cases} \phi^\top X_\tau^{-1} \sum_{i \geq 1} \phi_{\tau,i} V_{\tau+1}(\tilde{s}_{\tau+1,i}) + 32\sqrt{\frac{md\log\left(\frac{dH}{\epsilon\delta}\right)}{N}} + \frac{32md\sqrt{f_1}\log\left(\frac{dH}{\epsilon\delta}\right)}{N}, & \phi^\top \check{\Lambda}_\tau^{-1}\phi \leq 1; \\ 0, & \text{else} \end{cases} \tag{52}$$

    **end for**

    **for** $s \in \{\tilde{s}_{\tau-1,i}\}_{i \geq 1}$ **do**

        $V_\tau(s) = \text{Range}_{[0,1]}(\max_a Q_\tau(s,a))$;

        $\pi_\tau(s) = \arg\max_a Q_\tau(s,a)$;

    **end for**

**end for**

$V_0 \leftarrow \frac{1}{N}\sum_{i=1}^{N} V_1(s_{1,i}) + 2\sqrt{\frac{H^2\log(1/\delta)}{N}} + 2\frac{H\log(1/\delta)}{N}$;

**return:** $\{V_0, \pi\}$

---

**Algorithm 6** Policy-Execution

---

1: **Input** $h, \{\pi^{i,h}\}_{i=1}^{m}, \check{\Lambda}_h$ :

2: $\pi \leftarrow \text{uniform}(\{\pi^{i,h}\}_{i=1}^{m})$;

3: **for** $\tau = 1, 2, \ldots, H$ **do**

4:     **for** $z = 1, 2, \ldots, 2m^2 + 1$ **do**

5:         **for** $j = 1, 2, \ldots, N$ **do**

6:             Run $\pi$ to observe the feature $\phi_{h,j}$ and the next state $\tilde{s}_{h,j}$;

7:             $\lambda_{h,j} \leftarrow \min\left\{\sqrt{\frac{f_1}{\phi_{h,j}^\top \check{\Lambda}_h^{-1} \phi_{h,j}}}, 1\right\}$;

8:         **end for**

9:         $\mathcal{D}_h^\tau(z) \leftarrow \{\phi_{h,j}, \tilde{s}_{h,j}, \lambda_{h,j}\}_{j=1}^{N}$;

10:     **end for**

11:     $\mathcal{D}_h^\tau \leftarrow \{\mathcal{D}_h^\tau(z)\}_{z=1}^{2m^2+1}$

12: **end for**

13: **return** : $\mathcal{D}_h \leftarrow \{\mathcal{D}_h^\tau\}_{\tau=1}^{H}$.

---

---

**Algorithm 7** Ini-Sampling

---

1: **Initialization:** $\Lambda_0 \leftarrow \mathbf{I}$, $K \leftarrow 20d \log(1/\upsilon)$, $n \leftarrow 1600 \frac{d^2 H}{\epsilon}$, $f_2 \leftarrow \frac{1600 dH}{3\epsilon}$;
2: **for** $\ell = 1, 2, \ldots, K$ **do**
3:     $F \leftarrow \mathbf{0}$;
4:     **for** $i = 1, 2, \ldots, n$ **do**
5:         Play the local optimal design policy $\pi_{\mathtt{G}}(\cdot)$, observe the feature $\phi_{i,\ell}$;
6:         $F \leftarrow F + \mathtt{T}(\phi_{i,\ell}\phi_{i,\ell}^\top, f_2\Lambda_{\ell-1})$
7:     **end for**
8:     **if** $\frac{F}{n} + \upsilon\mathbf{I} \succeq \frac{1}{2}\Lambda_{\ell-1}$ **then**
9:         $\check{\Lambda}_1 \leftarrow F + 2n\upsilon\mathbf{I}$ and **break**;
10:     **else**
11:         $\Lambda_\ell \leftarrow F/n$;
12:     **end if**
13: **end for**
14: **for** $h = 1, 2, \ldots, H$ **do**
15:     **for** $i = 1, 2, \ldots, 2m+1$ **do**
16:         **for** $j = 1, 2, \ldots, N$ **do**
17:             Play the local optimal design policy $\pi_{\mathtt{G}}(\cdot)$, observe initial state $s_{1,j}^h(i)$, feature $\phi_{1,j}^h(i)$
               and the next state $\tilde{s}_{1,j}^h(i) = s_{2,j}^h(i)$;
18:             $\lambda_{1,j}^h(i) \leftarrow \min\left\{\sqrt{\frac{f_1}{(\phi_{1,j}^h(i))^\top \check{\Lambda}_1^{-1}(\phi_{1,j}^h(i))}}, 1\right\}$
19:         **end for**
20:         $\mathcal{D}_0^h(i) \leftarrow \{s_{1,j}^h(i)\}_{j=1}^N$
21:         $\mathcal{D}_1^h(i) \leftarrow \{\phi_{1,j}^h(i), \tilde{s}_{1,j}^h(i), \lambda_{1,j}^h(i)\}_{j=1}^N$;
22:     **end for**
23:     $\mathcal{D}_0^h \leftarrow \{\mathcal{D}_0^h(i)\}_{i=1}^{2m+1}$;
24:     $\mathcal{D}_1^h \leftarrow \{\mathcal{D}_1^h(i)\}_{i=1}^{2m+1}$
25: **end for**
26: **return:** $\{\mathcal{D}_0^h, \mathcal{D}_1^h\}_{h=1}^H$;

---

**Lemma 17.** *Recall the definition of $\check{\Lambda}_1$ in Algorithm 7. With probability $1 - \frac{\delta}{2H}$, for any sub-dataset of Algorithm 1 for the $h$-th layer $\{\phi_{h,i}, \tilde{s}_{h,i}, \lambda_{h,i}\}_{i \in [N]}$, it holds that*

$$\max_\pi \Pr_\pi \left[\phi_1^\top \check{\Lambda}_1 \phi_1 > 1\right] \leq \frac{\epsilon}{8H^2};$$

$$\sum_{i=1}^N \lambda_{1,i}^2 \phi_{1,i}\phi_{1,i}^\top + z\mathbf{I} \succeq \frac{N}{8m}\check{\Lambda}_1;$$

$$\lambda_{1,i}^2 \phi_{1,i}^\top \check{\Lambda}_1 \phi_{1,i} \leq 1, \forall i \in [N].$$

*Proof.* The third inequality follows by definition of $\lambda_{1,i}$. It suffices to prove the first two inequalities.

**The first condition.** Define $F(\Lambda) = \mathbb{E}_{s \sim d_{\mathrm{ini}}, a \sim \pi_{\mathtt{G}}(s)} \left[\mathtt{T}(\phi_1\phi_1^\top, f_2\Lambda)\right]$. Then $F(\Lambda)$ is non-increasing in $\Lambda$. Fix $\Lambda$. Let $\{\phi_i\}_{i=1}^n$ be the feature vectors by running $\pi_{\mathtt{G}}(\cdot)$ for $n$ rounds. By Lemma 9, with probability $1 - \frac{\delta}{32HK}$,

$$\hat{F}(\Lambda, n) := \sum_{i=1}^n \mathtt{T}\left(\phi_i\phi_i^\top, f_2\Lambda\right) \succeq nF(\Lambda) - 4\sqrt{n}\log(HKd/\delta)\Lambda \succeq nF(\Lambda) - \frac{n}{10d}\Lambda. \tag{53}$$

In a similar way, with probability $1 - \frac{\delta}{32HK}$,

$$\hat{F}(\Lambda, n) \preceq nF(\Lambda) + \frac{n}{10d}\Lambda. \tag{54}$$

Recall the definition of $\{\Lambda_\ell\}_{\ell \geq 0}$ in Algorithm 7. Assume the break condition in line 9 is not triggered in the first $\tau$ rounds. By equation 53 and equation 54, with probability $1 - \frac{\delta}{32H}$, for all $1 \leq \ell \leq \tau$,

$$\Lambda_\ell + \upsilon \mathbf{I} \not\succeq \frac{1}{2}\Lambda_{\ell-1};$$

$$\Lambda_\ell \preceq F(\Lambda_{\ell-1}) + \frac{1}{10d}\Lambda_{\ell-1} \preceq (1 + \frac{1}{10d})\Lambda_{\ell-1};$$

$$\Lambda_\ell \succeq F(\Lambda_{\ell-1}) - \frac{1}{10d}\Lambda_{\ell-1}. \tag{55}$$

Let $\tilde{\Lambda}_\ell = \Lambda_\ell + 2\upsilon\mathbf{I}$. It then follows that

$$\tilde{\Lambda}_\ell \not\succeq \frac{1}{2}\tilde{\Lambda}_{\ell-1}$$

$$\tilde{\Lambda}_\ell \preceq (1 + \frac{1}{10d})\tilde{\Lambda}_{\ell-1};$$

$$\tilde{\Lambda}_\ell \succeq 2\upsilon\mathbf{I}.$$

As a result, we have that

$$\log(\det(\tilde{\Lambda}_\ell)) - \log(\delta(\tilde{\Lambda}_{\ell-1})) \leq -\log(2) + d\log(1 + \frac{1}{10d}) \leq -0.1, \tag{56}$$

which implies that

$$d\log(\upsilon) \leq \log(\det(\tilde{\Lambda}_\tau)) \leq -0.1\tau \tag{57}$$

and $\tau \leq 10d\log(1/\upsilon)$. Therefore, the break condition in line 9 will be triggered within $K$ rounds.

Now we verify the first inequality. By definition, there exists some $\ell$ such that $\frac{\hat{F}(\Lambda_{\ell-1},n)}{n} + \upsilon\mathbf{I} \succeq \frac{1}{2}\Lambda_{\ell-1}$ and $\check{\Lambda}_1 = \hat{F}(\Lambda_{\ell-1},n) + 2n\upsilon\mathbf{I}$, which means that

$$F(\Lambda_{\ell-1}) + \frac{1}{10d}\Lambda_{\ell-1} \succeq \hat{F}(\Lambda_{\ell-1},n)/n \succeq \frac{1}{2}\Lambda_{\ell-1} - \upsilon\mathbf{I}.$$

As a result, we learn that

$$F(\Lambda_{\ell-1}) + \upsilon\mathbf{I} \geq \left(\frac{1}{2} - \frac{1}{10d}\right)\Lambda_{\ell-1}$$

and

$$\frac{\hat{F}(\Lambda_{\ell-1},n)}{n} + \upsilon\mathbf{I} \succeq F(\Lambda_{\ell-1}) + \upsilon - \frac{1}{10d}\Lambda_{\ell-1} \succeq \left(\frac{1}{2} - \frac{1}{5d}\right)\Lambda_{\ell-1}. \tag{58}$$

Continuing the computation we have that

$$\max_\pi \mathrm{Pr}_{s \sim d_{\mathrm{ini}}, \pi}\left[\phi_1^\top \check{\Lambda}_1^{-1}\phi_1 \geq 1\right] \leq \mathrm{Pr}_{s \sim d_{\mathrm{ini}}, \pi_{\mathsf{G}}}\left[\mathrm{Trace}\left(\mathbb{E}_{\phi \sim \pi_{\mathsf{G}}(s)}[\phi\phi^\top]\check{\Lambda}_1^{-1}\right) \geq 1/d\right]$$

$$\leq d\mathrm{Pr}_{s \sim d_{\mathrm{ini}}, \pi_{\mathsf{G}}}\left[\phi_1^\top \check{\Lambda}_1^{-1}\phi_1 \geq 1/d\right]$$

$$\leq d\mathrm{Pr}_{s \sim d_{\mathrm{ini}}, \pi_{\mathsf{G}}}\left[\phi_1^\top \left(\frac{\hat{F}(\Lambda_{\ell-1},n)}{n} + 2\upsilon\mathbf{I}\right)^{-1}\phi_1 \geq n/d\right]$$

$$\leq d\mathrm{Pr}_{s \sim d_{\mathrm{ini}}, \pi_{\mathsf{G}}}\left[\phi_1^\top \Lambda_{\ell-1}^{-1}\phi_1 \geq \frac{n}{3d}\right]. \tag{59}$$

Continuing the computation, we have that

$$d\mathrm{Pr}_{s \sim d_{\mathrm{ini}}, \pi_{\mathsf{G}}}\left[\phi_1^\top \Lambda_{\ell-1}^{-1}\phi_1 \geq \frac{n}{3d}\right] \leq \frac{3d}{n} \cdot \mathbb{E}_{s \sim d_{\mathrm{ini}}, \pi_{\mathsf{G}}}\left[\mathrm{Trace}\left(\mathsf{T}\left(\phi_1\phi_1^\top, \frac{n}{3d}\Lambda_{\ell-1}\right) \cdot (F(\Lambda_{\ell-1}))^{-1}\right)\right]$$

$$= \frac{3d}{n} \cdot \mathbb{E}_{s \sim d_{\mathrm{ini}}, \pi_{\mathsf{G}}}\left[\mathrm{Trace}\left(\mathsf{T}\left(\phi_1\phi_1^\top, f_2\Lambda_{\ell-1}\right) \cdot (F(\Lambda_{\ell-1}))^{-1}\right)\right]$$

$$= \frac{3d^2}{n}. \tag{60}$$

Here the second inequality is by the fact that $\frac{n}{3d} = f_2$. Therefore, we have that

$$\max_\pi \mathrm{Pr}_{s \sim d_{\mathrm{ini}}, \pi}\left[\phi_1^\top \check{\Lambda}_1^{-1}\phi_1 \geq 1\right] \leq \frac{3d^2}{n} \leq \frac{\epsilon}{8H}.$$

**The second condition.** Recall that $\check{\Lambda}_1 = \hat{F}(\Lambda_{\ell-1}, n) + 2nv\mathbf{I} \preceq n(1 + \frac{1}{10d})F(\Lambda_{\ell-1}) + 2nv\mathbf{I}$ and $\check{\Lambda}_1 \succeq \frac{n}{3}\Lambda_{\ell-1}$. Let $\{\phi_{1,j}, \tilde{s}_{1,j}, \lambda_{1,j}\}_{j=1}^N$ be a sub-dataset collected following line 16 to line 21 in Algorithm 7. Then we have that

$$
\mathbb{E}\left[\lambda_{1,j}^2 \phi_{1,j}\phi_{1,j}^\top\right] = \mathbb{E}_{s\sim d_{\mathrm{ini}}, \pi_{\mathsf{G}}}\left[\mathtt{T}(\phi_1\phi_1^\top, f_1\check{\Lambda}_1)\right] \succeq \mathbb{E}_{s\sim d_{\mathrm{ini}}, \pi_{\mathsf{G}}}\left[\mathtt{T}(\phi_1\phi_1^\top, f_2\Lambda_{\ell-1})\right] = F(\Lambda_{\ell-1}).
$$
(61)

Using Lemma 9, with probability $1 - \frac{\delta}{16m^2H^2}$, it holds that

$$
\sum_{j=1}^N \lambda_{1,j}^2 \phi_{1,j}\phi_{1,j}^\top \succeq NF(\Lambda_{\ell-1}) - (4\sqrt{N\log(dHm/\delta)} + 2\log(dHm/\delta)) \cdot f_1\check{\Lambda}_1
$$

$$
\succeq NF(\Lambda_{\ell-1}) - 6\sqrt{N\log(dHm/\delta)}f_1 \cdot (2nF(\Lambda_{\ell-1}) + 2nv\mathbf{I})
$$

$$
\succeq \frac{N}{2}F(\Lambda_{\ell-1}) - 12\sqrt{N\log(dHm/\delta)}f_1 \cdot 2nv\mathbf{I}
$$

$$
\succeq \frac{N}{4n}\check{\Lambda}_1 - (12\sqrt{N\log(dHm/\delta)}f_1 \cdot 2n + 4N)v\mathbf{I}
$$

$$
\succeq \frac{N}{8m}\check{\Lambda}_1 - z\mathbf{I}.
$$

The proof is completed.

$\square$

