# OpenReview forum: "Deployment Efficient Reward-Free Exploration with Linear Function Approximation"
_ICLR.cc/2025/Conference — Submitted to ICLR 2025_

### Official Review · Reviewer_pxSY · 2024-10-16

**Soundness:** 3
**Presentation:** 2
**Contribution:** 3
**Rating:** 6
**Confidence:** 3

**Summary:**

This paper studies deployment-efficient RL in linear MDPs. The authors contribute a reward-free exploration algorithm and prove it only takes $O(H)$ deployments to learn a near-optimal policy. That result does not rely on additional reachability assumption, and therefore, close the gap between the $\Omega(H)$ lower bound established in previous work.

**Strengths:**

The deployment efficient setting indeed reflects practical concerns and it is important.

The algorithm proposed in this paper has non-trivial and interesting ingredients.

The discussion in Section 4 is detailed and explain the insights clearly.

**Weaknesses:**

1. My main criticism would be the paper writing. Especially, the proof sketch in Section 6 does not provide too much information to me. It involves a lot notations and the authors did not explain the insights of the key steps intuitively. For example, in the proof sketch for Lemma 6, it is just mentioned which lemmas in appendix are used. Given the page limit has not been reached yet, It would be better to explain the intuition with more details.

    Besides, I'm not sure whether it is a good idea, but it may be easier for reader to understand if the authors can re-organize Section 4-6 to make connections among the explanation in Section 4, the algorithm design in Section 5 and the technique lemma in Section 6. Now the authors discuss them separately, which makes it hard to track which part of the algorithms correspond to which technique innovation.

2. I also have a few questions about the proofs. Please check Question part.

3. There are also quite a few typos. For example, line 350, 465, 475, the sub-scription of $\phi$. Besides, some notations in the appendix seem not clear to me. See Question part.

4. Given there are a lot notations, it would be better to have a table of frequently used notations in appendix for reference.

**Questions:**

1. Lemma 7, what do you mean by "maps a context $X$ to a distribution over $X$"? Why not just consider $\pi^G = \mathcal{K}$? Or it should be "context $x$"?

2. Lemma 8, can you explain why the last inequality in Eq.(10) holds?

3. Lemma 11 and 14, what do the $\mu$ and $\theta(v)$ denote, and how should we interpret them?

4. Lemma 15, it says "$\hat{F}\_0(\Lambda)$ be the value computed by line 12-24 of Algorithm 3". However, in 12-24 of Algorithm 3, $\hat{F}$ is defined on $\tilde{s}\_{h-1,i}$, which one of them do you refer to?

---

> ### Author Response · Authors · 2024-11-22
> **Response to Reviewer pxSY**
>
> We thank the reviewer for the detailed comments and interesting questions. Below is our response.
>
> **About the writings:** Thanks for the suggestion. We present the major high-level ideas in Section 4 and Section 5. We will try to combine the two sections together in the next version.
>
> **About the typos:** Thanks for the careful review. We have fixed the typos accordingly. We have also added a table of notations in the revision.
>
>
>
> **About the question in Lemma 7:** By "maps a context $X$ to a distribution over $X$", we mean that: given $X=\\{x_1,x_2,...,x_n\\}$, $\pi(X)$ is a distribution over $\\{x_1,x_2,...,x_n\\}$. In other words, $\pi(X)$ could also be viewed as a $n$-dimensional probability vector.
> We do not use $\mathcal{K}(X)$ because the computation cost for $\mathcal{K}(X)$ might be large, while an approximation could be efficiently computed.
>
> **About Eq.(10) in Lemma 8:**
> Note that  $\mathtt{T}(\phi\phi^{\top},\Lambda) = \min\\{\frac{1}{\phi^{\top}\Lambda^{-1}\phi},1  \\} \phi\phi^{\top}$.
> By the fact that $\mathrm{Trace}( \mathtt{T}(\phi\phi^{\top},\Lambda)\cdot (\Lambda)^{-1} ) = \mathrm{Trace}(  \min\\{\frac{1}{\phi^{\top}\Lambda^{-1}\phi},1\\} \phi\phi^{\top}\cdot \Lambda^{-1}  )$, which is equal to $\min\\{   \phi^{\top}\Lambda^{-1} \phi , 1\\}$. Then the inequality holds by the condition (6).
>
>
> **About $\mu$ and $\theta(v)$ in Lemma 11 and 14:** $\mu$ is the underlying unknown transition kernel, and $\theta(v)$ is the vector $\mu^{\top}v$, which is the payoff vector with value function as v. Here we omit the script of $h$ for simplicity.
>
> **About $F^0(\Lambda)$ in Lemma 15:**  We are sorry for the ambiguity. Here $F^0(\Lambda)$ denotes the matrix in the return line. We have fixed accordingly.

---

> > ### Comment · Reviewer_pxSY · 2024-11-22
> >
> > Thanks for the responses. I do not have further questions and I will keep my score.

---

### Official Review · Reviewer_dGaZ · 2024-11-01

**Soundness:** 3
**Presentation:** 2
**Contribution:** 2
**Rating:** 5
**Confidence:** 3

**Summary:**

The paper studies deployment efficient reward-free exploration with linear function approximation. The proposed algorithm achieves nearly optimal deployment efficienc and does not depend on the reachability coefficient or the explorability coefficient of the underlying MDP as previous literature.

**Strengths:**

The paper is clearly written. The algorithm design and theoretical analysis is non-trivial, especially the idea of quantification/design of the truncated state-action pairs in the underling MDP.

**Weaknesses:**

The algorithm is heavily engineered with artifical tricks, which makes the algorithms not tractable (correct me if I am wrong). While the initial goal was to inspire practical algorithms that enhance deployment efficiency in RL applications, one might question how effectively this approach serves that purpose.

**Questions:**

1. Can you provide some discussions on the tractability of the algorithm? Can the current algorithm motivate some practical algorithms? Is any experiment possible to show the performance?

2. The algorithm required $2m^2+1$ independent datasets, and $m$ is set to be $32000d^2H^3/\epsilon$, which is insanely large. This requirement alone could make the 'deployment effciency' vacuous. I'd like some explanation on this point.

---

> ### Author Response · Authors · 2024-11-22
> **Response to Reviewer dGaZ**
>
> We thank the reviewer for the detailed comments. We present our response as below.
>
> **About tractability:** The current algorithm suffers from a high polynomial sample size, which is unaffordable in most applications.
>
> **About repeated sampling:** We admit the current algorithm is not practical in many cases. We have to use these independent datasets to keep the correctness of the linear regression. It would be an interesting direction to weaken the dependence to reduce the sample complexity.

---

### Official Review · Reviewer_H8jv · 2024-11-04

**Soundness:** 2
**Presentation:** 1
**Contribution:** 2
**Rating:** 3
**Confidence:** 1

**Summary:**

This paper studies deployment efficient reward free exploration problem in linear MDPs. They proposed an algorithm which achieved optimal deployment complexity $O(H)$ and polynomial sample complexity while no assumption on the reachability of the underlying MDP is needed. This result provides a positive answer to the open problem whether people can get rid of the reachability assumption under the linear MDPs setting.

**Strengths:**

1. The problem of studying deployment-efficient exploration policy is well-motivated both from a practical and theoretical perspective.

**Weaknesses:**

I admit that I cannot assess the correctness of their method so I will set my confidence score as 1. However, I hold the opinion that even though assuming the theory is correct, I don't think the current version is ready to be published. The presentation and organization of this paper need to be improved a lot. It is really hard to understand their idea even qualitatively. For example, Section 4 should be the place where they explain their high-level idea, however, after reading this section, people still have no idea on what exact technical problem will emerge if there is no assumption on reachability and how they handle these technical problems.

**Questions:**

Can the authors briefly explain what exact technical problem will emerge if there is no assumption on reachability (by some lines of equations) and how they handle these technical issues?

---

> ### Author Response · Authors · 2024-11-22
> **Response to Reviewer H8jv**
>
> We thank the reviewer for the efforts in this review.  We response as follows.
>
>
>
> **About technical problem without reachability :**
>
>  The main problem is about evaluation of $\Lambda_h(\pi):=\mathbb{E}_{\pi}[\phi_h\phi_h^{\top]}]$ with the offline dataset before the $h$-th layer.
> We remark that online estimation is impossible in our setting because of the limitation of deployments. In this offline value evaluation (OPE) problem, with previous algorithm, one could get $\hat{\Lambda}$ be such that $\\|\hat{\Lambda}-\Lambda_h(\pi)\\| = O(\mathrm{err}_h)$, where $\mathrm{err}_h$ depends on the coverage ratio of datasets before the $h$-th layer.
>
> Note the iteration method to find the coverage information matrix for the $h$-th layer is as follows: $\Lambda_1 = \zeta I$, $\pi_1$ be the empirical optimal policy w.r.t. $r_1 =  \min\\{\phi_h^{\top}\Lambda_1^{-1}\phi_h,1\\}$, and then  update $\Lambda_2 = \Lambda_1 + \hat{\Lambda}(\pi)$. Here we use the above OPE method to compute an estimator $\hat{\Lambda}(\pi)$ for $\Lambda_h(\pi)$, which has error $\mathrm{err}_h I$.
>
> We do this recursively until we find some $n$ such that $\max_{\pi}E[r_n]\leq O(\epsilon)$. In general, we require $n = \text{poly}(d/\epsilon)$. In this process, the offline error $\mathrm{err}_h I$ is accumulated to be $n  \cdot \mathrm{err}_h I$, and will leads to $\sqrt{n \cdot \mathrm{err}_h}$ error for the $h+1$-th layer.
>
> In words, $\mathrm{err}_{h+1}\geq \Omega( \sqrt{ n\cdot \mathrm{err}_h}) $ for some $n\geq 1$. As a consequence, the error increase very fast as the horizon gets large. We also remark that, with the reachability assumption, there exists $\pi$ such that $\Lambda_h(\pi)\geq z\mathbf{I} $ for some parameter $z$, which could help remove the regularization term by playing $\text{poly}(dH/z)$ samples.
>
>
>
> With the proposed clipped evaluation method, instead of bounding $\hat{\Lambda}-\Lambda_h(\pi)$ with $\mathrm{err}_h \mathbf{I}$, we manage to prove that $\hat{\Lambda}-\Lambda_h(\pi)$ is bounded by $\mathrm{err}_h\Lambda$, where $\Lambda$ is some proper upper bound. One can thing $\Lambda$ as $\Lambda_h(\pi)$. In this case, we have $(1-\mathrm{err}_h)\Lambda_h(\pi)\leq \hat{\Lambda}\leq (1+\mathrm{err}_h)\Lambda_h(\pi)$, which help to eliminate the term $\mathrm{err}_h \mathbf{I}$ in the next steps.

---

### Official Review · Reviewer_cBGc · 2024-11-05

**Soundness:** 3
**Presentation:** 2
**Contribution:** 2
**Rating:** 5
**Confidence:** 3

**Summary:**

This paper investigates deployment-efficient reward-free exploration with linear function approximation in reinforcement learning. The goal is to explore a linear Markov Decision Process (MDP) without revealing the reward function, while minimizing the number of exploration policies. A reinforcement learning algorithm is proposed with polynomial sample complexity, achieving near-optimal deployment efficiency for linear MDPs.

**Strengths:**

1. The paper provides a thorough and precise theoretical justification for the proposed algorithm.

2. The proposed algorithm demonstrates a promising sample complexity bound that is polynomial in both the feature dimension and horizon length.

**Weaknesses:**

1. While the algorithm is theoretically sound, the paper does not provide a detailed comparison with other existing methods in terms of empirical performance.

2. The paper introduces a reward-free exploration strategy but does not provide an in-depth analysis of the exploration-exploitation trade-off inherent in the algorithm. Specifically, it lacks a clear explanation of how the exploration policy adapts over time, especially in the presence of uncertainties in the reward-free setting.

**Questions:**

1. How does the proposed algorithm handle cases where the linear function approximation is not perfect or when the feature dimension is large? What are the potential implications for the performance bounds under these circumstances?

2. The exploration strategy in the proposed algorithm minimizes exploration policies, but how does the method balance this trade-off with the inherent uncertainty in large-scale environments? Are there conditions where minimizing the exploration policies might hinder the learning process in practice?

3. How does the proposed reward-free exploration strategy balance the exploration of state-action pairs with the need to minimize exploration policies?

---

> ### Author Response · Authors · 2024-11-22
> **Response to Reviewer cBGc**
>
> We thank the reviewer for careful review and interesting questions. Below we present our response.
>
>
>
> **About numerical experiments:**  Due to repeated sampling to keep data independence, the order of $d$ and $H$ is large in the final sample complexity, which is a major obstacle to implement this algorithm with affordable number of samples. It is an interesting direction to reduce the order of these factors.
>
> **About price of  reward-free learning:** The algorithm is naturally reward-free, since the process of collecting data is independent of reward. As a price, the sample complexity is comparably large since we need to sample the same level repeatedly for many times.
>
> **About linear inaccuracy:** When linear approximation is not accurate, the algorithm suffers linearly from the error term. By assuming  $|Pv-\phi^{\top}\theta(v)|_{\infty}\leq \xi$ for all $v$ with infinite norm at most $1$, we could conduct the clipped offline evaluation with
> linear regression so that error due to linear inaccuracy is at most poly$(dH\xi) V$. To prove this claim, we provide a modified version of Lemma 14 allowing linear inaccuracy $\xi$ by adding poly$(dH\xi)V$ to the right hand side of the two inequalities.
>
> **About large d :** As presented in the paper, the sample complexity depends on a  polynomial factor of $d$, which is necessary in the worst case. In the case $d$ is very large, we require additional assumptions to simplify the function class so that the MDP is learnable.
>
> **About environments with inherent uncertainty:** In an environment with inherent uncertainty, there would be a lower bound for the number of re-deployments. That is, it is impossible to learn the optimal policy with limited deployments. Nevertheless, our work provides ideas to explore more adaptive algorithms. For example, the proposed algorithm is robust under linear inaccuracy.
>
> **About balance between the exploration  and the need to minimize exploration policies:** Without the limitation on deployments, we could use a more adaptive strategy to explore the environment. For example, we can design reward function adaptively to explore some certain subsets of the state-action space with a set of different policies. However, in the case the number of deployments is limited, we have to find one policy to explore the whole state-action space, and we do so by devising efficient offline policy evaluation and offline policy optimization algorithms.

---

> > ### Comment · Reviewer_cBGc · 2024-11-26
> >
> > Thanks for the responses. I will keep my score.

---

### Meta-Review · Area_Chair_xzy5 · 2024-12-19

**Metareview:**

Deployment Efficient Reward-Free Exploration with Linear Function Approximation

Summary: The paper proposes a new reward-free exploration algorithm for reinforcement learning (RL) with linear function approximation that optimizes deployment efficiency without assuming reward availability during exploration. The method minimizes the number of exploration policies during deployments while achieving polynomial sample complexity in feature dimensions and horizon lengths, overcoming challenges posed by restrictive assumptions like reachability and explorability in prior approaches. By truncating state-action pairs based on data and employing offline policy evaluation and optimization, the algorithm ensures robust performance and avoids reliance on reachability coefficients, which can be arbitrarily small in some MDPs.

Comments: We received four reviews, with the scores 3, 5, 5, 6. One of the reviewers has provided a very short review and has indicated that they are of low confidence about the feedback. Excluding this review, this paper has an average score of 5.33.

Reviewers have given positive comments about some aspects of the paper. However, the reviewers have raised multiple concerns about the paper which outweigh the positive aspects. Reviewer cBGc has asked about the balance between exploration-exploitation trade-off which is not discussed in this paper. Reviewer dGaZ has pointed out that the proposed algorithm required $2m^2+1$ independent datasets, and $m$ is set to be $32000d^2H^3/\epsilon$, which is very large, and this defeats the claim about the 'deployment efficiency'. Reviewer pxSY has commented about the quality of the presentation as it is very difficult to follow the proof ideas and insights presented in Section 4 and Section 6. This reviewer has also pointed out ambiguities in the proof.

In summary, while the theoretical contributions are commendable,  the paper falls short in terms of clarity, empirical validation, and practical relevance. I recommend the authors to address these concerns in a resubmission.

**Additional Comments On Reviewer Discussion:**

Please see the "Comments" in the meta-review.

---

### Decision · Program_Chairs · 2025-01-22

Reject